# Pre-training Limited Memory Language Models with Internal and External Knowledge

**Linxi Zhao, Sofian Zalouk, Christian K. Belardi, Justin Lovelace, Jin Peng Zhou**
**Ryan Thomas Noonan, Dongyoung Go**
**Kilian Q. Weinberger, Yoav Artzi, Jennifer J. Sun**
Department of Computer Science
Cornell University
{lz586, saz43, ckb73, jl3353, jz563, rtn27, dg793}@cornell.edu
{kilian, jjs533}@cornell.edu   yoav@cs.cornell.edu

## Abstract

Neural language models are black-boxes – both linguistic patterns and factual knowledge are distributed across billions of opaque parameters. This entangled encoding makes it difficult to reliably inspect, verify, or update specific facts. We introduce Limited Memory Language Models (LmLm)[†] a new class of language models that externalizes factual knowledge to external database during pre-training rather than memorizing them. Our pre-training approach strategically masks externally retrieved factual values from the training loss, thereby teaching the model to perform targeted lookups rather than relying on memorization in model weights. Our experiments demonstrate that LmLms achieve competitive performance compared to significantly larger LLMs on standard benchmarks, while offering the advantages of explicit, editable, and verifiable knowledge bases.

## 1 Introduction

Many challenges with deploying LLMs in real-world applications originate from the fact that training on vast text corpora exposes LLMs to vast amounts of knowledge (Kaddour et al., 2023; Allen-Zhu & Li, 2024a). Ideally, the knowledge in language models should be fully controllable—a customer service agent for a restaurant chain shouldn't have the ability to answer questions about historical facts, prescription medicine, real estate law, etc. Unfortunately, current LLM pre-training procedures entangle the learning of knowledge with language competency in their neural weights. This has significant implications for both training and inference. During pre-training, facts must be observed many times for reliable memorization (Allen-Zhu & Li, 2024c; Kandpal et al., 2023). During inference, it is difficult to unlearn knowledge that may be outdated or inappropriate for a particular application (Longpre et al., 2021; Maini et al., 2024; Ovadia et al., 2024; Deeb & Roger, 2025). This tight coupling of knowledge and linguistic ability makes updating one without affecting the other extremely challenging.

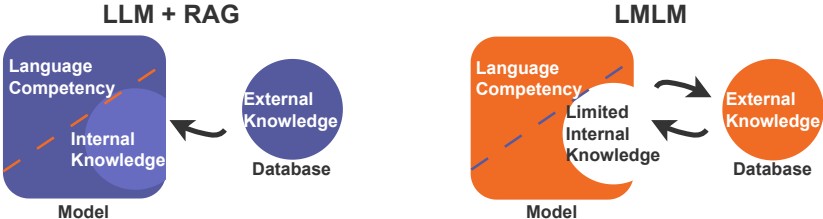

Figure 1: A schematic illustration of LmLm. Unlike RAG (left), which exclusively *adds* knowledge from external sources, LmLm *offloads* knowledge from the LLMs to the external database during pre-training.

---

[†]LmLm stands for **L**imited **M**emory **L**anguage **M**odel and is pronounced "LamLam".

[‡]We open-source our code and models at https://github.com/kilian-group/LMLM.

Our vision is a pre-trained language model where knowledge is fully modular and controllable, and can be added or deleted easily for different use cases. This leads us to conclude that storing knowledge inside the model weights during pre-training should be limited as much as possible. This gives rise to the following research question:

***Can factual memorization be disentangled from language understanding in language models?***

To realize this vision, we introduce LIMITED MEMORY LANGUAGE MODELS (LMLM), a new class of language models with a pre-training recipe that teaches the model to query an external database for entity-level facts rather than memorizing them. We provide an integrated solution for LMLMs spanning data preparation, pre-training, and inference. To prepare the training data, we annotate the pre-training corpus with database lookups to offload factual content, using a small, cost-effective LM fine-tuned for this task. During pre-training, the returned facts are *masked from the loss*, systematically separating factual knowledge from the neural weights. During inference, instead of recalling memorized facts, the model *queries the database*.

Unlike the dominant paradigm of post-training and inference-time approaches such as retrieval-augmented generation (RAG) (Lewis et al., 2021) that *maximize* access to external information, our approach takes the perspective of *limiting* knowledge stored inside the model parameters during *pre-training* (Figure 1). This frees up model capacity, disentangles knowledge from language competency, and enables precise control over factual knowledge. During *inference*, LMLM is naturally compatible with RAG systems, tool calling (Schick et al., 2023), or task-specific fine-tuning (Wei et al., 2022). LMLMs always rely on lookups for factual information, so attempts to access out-of-scope knowledge trigger detectable lookup failures. In contrast, traditional RAG models without LMLM pre-training typically fall back to their *internal knowledge*, which is beyond the control of the user and may lead to hallucinations or misinformation (Longpre et al., 2021; Mallen et al., 2023; Sun et al., 2025).

We compare our LMLMs against LLM counterparts of the same size, pre-trained on the same corpus without external memory. Our experiments demonstrate that LMLMs achieve lower perplexity than comparable LLMs (§ 4.2). Crucially, by externalizing knowledge from model weights, LMLMs achieve significant parameter efficiency—our 382M-parameter model matches the factual precision of a LLAMA2-7B (§ 4.4). Knowledge externalization also enables instant, verifiable updates and unlearning through simple database operations (§ 4.3). Additionally, we present findings regarding the knowledge decoupling achieved by LMLM and examine the trade-off between internal memorization and external offloading (§ 5). LMLMs offer a pathway toward language models with substantially reduced dependency on parameter count for factual accuracy. LMLMs have potential for integration with knowledge representation (Pan et al., 2024), symbolic reasoning (Chaudhuri et al., 2021), and mechanistic interpretability (Bereska & Gavves, 2024), potentially transforming how language models store, access, and maintain knowledge.

## 2  RELATED WORK

**Parametric vs Non-Parametric Knowledge.**  Language models represent knowledge in two main ways: parametric, encoded in model weights during training, and non-parametric, retrieved from external sources during pretraining or inference (Mallen et al., 2023). Pretrained models have been shown to function as implicit knowledge bases, storing millions of facts in parameters (Devlin et al., 2019; Petroni et al., 2019). Scaling model size and training data can further increase this

Table 1: Comparison of selected related work. ✓ = supported, ✗ = not supported. Full version in Table 13 in Appendix.

| Model | Knowledge Storage | | | Performance | | | Goal |
|---|---|---|---|---|---|---|---|
| | **Internal** | **External** | **Integration** | **PPL↓** | **Factuality↑** | **Unlearning** | **Goal** |
| *Inference* | | | | | | | |
| kNN-LM | ✓ | DS | kNN search + prob. interp. | ✓ | ✓ | ✗ | Neighbor interpolation |
| RAG | ✓ | Docs | Retrieved docs in prompt | ✗ | ✓ | ✗ | Improve factuality |
| *Post-training* | | | | | | | |
| Toolformer | ✓ | APIs | Tool calls | ✗ | ✓ | ✗ | Extend via tools |
| *Pretraining* | | | | | | | |
| RETRO | ✓ | Docs | Cross-attn. to retrieved chunks | ✓ | ✓ | ✗ | Scale with retrieval |
| MemSinks | ✓ (sink neurons) | – | Memorization isolation | ✓ | ✗ | ✓ | Localize memorization |
| Memory[3] | ✓ (partial) | Sparse KV | Self-attention | ✓ | ✓ | ✗ | Reduce parametric memorization |
| **LMLM (ours)** | ✓ (partial) | KB | Explicit lookup | ✓ | ✓ | ✓ | **Decouple facts from weights** |

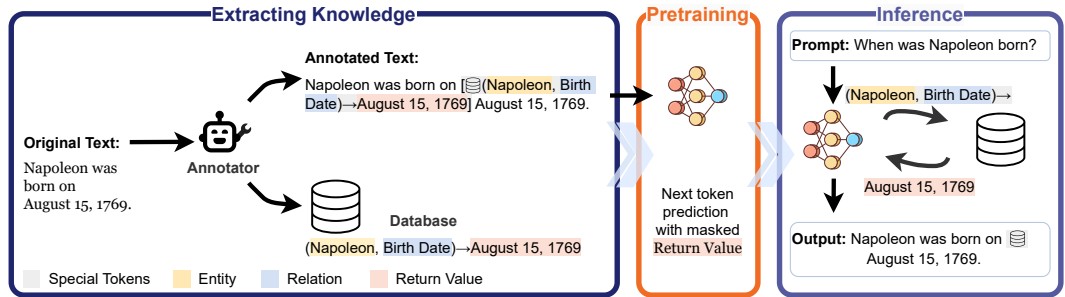

Figure 2: **Overview of the LMLM framework.** Our framework consists of *(Left) Data Preparation*, where entity-level facts are automatically annotated and stored in an external database; *(Middle) Pretraining*, where the model is trained on the annotated text while excluding return values from the loss to discourage memorization; and *(Right) Inference*, where the model interleaves text generation with database lookups to ground its outputs on retrieved facts.

knowledge capacity and improve factual precision (Brown et al., 2020; Izacard & Grave, 2022; Allen-Zhu & Li, 2024c). Yet, recent findings suggest that parametric knowledge may remain under-trained if the knowledge is observed less than a few hundred times during training (Allen-Zhu & Li, 2024c). Broadly, parametric knowledge suffers from well-documented issues, including hallucination, staleness, weak attribution, and limited adaptability (Peng et al., 2023a; Huang et al., 2024; Kandpal et al., 2023; Bi et al., 2025; Allen-Zhu & Li, 2024b), motivating the shift toward non-parametric approaches that explicitly externalize factual knowledge.

**Retrieval-Augmented LLMs.** A large body of work improves factuality by incorporating external knowledge at inference or training time. Retrieval-augmented generation (RAG) retrieves relevant passages (Lewis et al., 2021; Izacard & Grave, 2021; Shi et al., 2023), while tool-augmented models such as Toolformer extend model capabilities with API or tool calls (Schick et al., 2023; Yao et al., 2023). Retrieval-based pretraining approaches, including REALM (Guu et al., 2020) and RETRO (Borgeaud et al., 2022), demonstrate that integrating retrieval during training can reduce memorization and improve generalization. Semi-parametric models further externalize knowledge by attaching non-parametric key–value memories (Khandelwal et al., 2020; Peng et al., 2023b; Shi et al., 2022; Pan et al., 2023; Wu et al., 2022; Verga et al., 2021; Wang et al., 2024; Wei et al., 2025; Yang et al., 2024). In contrast, LMLM directly restricts memorization during pre-training. This design enables scalable and dynamic factual memory, and naturally supports properties such as instant unlearning—capabilities that are difficult to achieve with prior retrieval-augmented methods (see Table 1 and Table 13).

**Machine Unlearning.** Machine unlearning aims to remove specific knowledge from a trained language model without full retraining (Jang et al., 2022; Cao & Yang, 2015). Existing approaches mainly rely on gradient-based updates (e.g., GA, GD (Liu et al., 2022)) or preference-based optimization (e.g., IdkDPO, NPO, SimNPO (Maini et al., 2024; Zhang et al., 2024b; Fan et al., 2025; Rafailov et al., 2024)). While effective on unlearning targeted knowledge on TOFU benchmark (Maini et al., 2024), these methods remain difficult to scale to larger forget sets and often degrade utility or erase unintended related knowledge, reflecting the entanglement of factual content and general ability in model parameters. MemSinks (Ghosal et al., 2025) instead introduces mechanisms to isolate memorized content during training, making it easier to unlearn. Our approach departs from both lines by structurally decoupling factual knowledge from the model itself. Facts are externalized to a database, so forgetting reduces to deleting entries. This design enables scalable, precise, and verifiable unlearning, while naturally extending to knowledge editing tasks (Mitchell et al., 2022; Sinitsin et al., 2020; Fang et al., 2025).

## 3 LIMITED MEMORY LANGUAGE MODELS

LMLM is a new class of language models designed to offload factual knowledge to an external database rather than store it implicitly in model parameters *from the pre-training stage* (Figure 2). We outline pre-training data preparation (§ 3.1), pretraining with lookup masking and inference (§ 3.2).

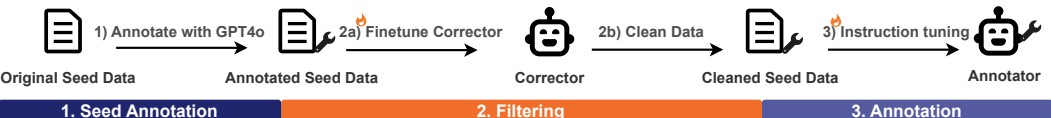

Figure 3: **Training the ANNOTATOR model.** We distill high-quality annotations from GPT-4o into a lightweight model that learns to identify and externalize factual knowledge from raw pre-training text, enabling scalable annotation of the full corpus.

### 3.1 PRETRAIN DATA PREPARATION: SEPARATING KNOWLEDGE

We begin by extracting atomic entity-level factual knowledge from pretrain corpus and constructing a compact external database. These extracted facts are then interleaved with the original pretraining corpus through explicit lookup calls.

**Knowledge Specification.** We focus primarily on *entity-level atomic factual knowledge*, a natural starting point within the broader LMLM framework. We define facts as triplets of the form: (`entity`, `relation`) → `value`. [1] This level of granularity aligns with previous definitions (Mallen et al., 2023) and represents the most compact and tractable form of factual knowledge to disentangle from the intertwined linguistic patterns in raw text. It also naturally maps to a knowledge graph structure, where triplets define nodes and edges (Choudhary & Reddy, 2023). These atomic facts are ideal for externalization: they are straightforward to extract and verify, yet hard to encode in the model parameters, making them well-suited for storage in an external database.

**Automating Knowledge Extraction.** Manually extracting factual triples and constructing knowledge graphs at the scale of pre-training data is a major challenge (Hu et al., 2024). We address this by first obtaining high-quality seed annotations with GPT-4o, then refining them through a filtering stage, and finally distilling the result into a lightweight ANNOTATOR that scalably externalizes factual knowledge from raw text (Figure 3). Additional details and prompts are provided in Appendix A.1.

1. *Seed annotation:* We use GPT-4o to annotate a small seed dataset of $M$ knowledge-intensive documents with lookup calls and return values. We use $M = 1000$ in our setting.
2. *Filtering:* We fine-tune a CORRECTOR model (LLAMA-3.1-8B-INSTRUCT) on the seed annotations with intentional underfitting. The CORRECTOR assigns high loss to improperly formatted, contextually unsupported, or overly specific lookup calls. We discard the top $10\%$ by loss.
3. *Annotation:* We apply instruction-tuning to an ANNOTATOR model (also LLAMA-3.1-8B-INSTRUCT) on the cleaned data and use it to annotate the full pre-training corpus at scale.

This annotation process serves two purposes: (1) *Database Construction:* The extracted triplets form a token-efficient external database that scales with the size of the pre-training corpus. (2) *Pre-training Corpus Generation:* Lookup calls are interleaved with the original text, enabling the model to learn when to rely on internal knowledge and when to issue a lookup.

### 3.2 PRE-TRAINING AND INFERENCE

**Pre-training** We adopt a standard next-token prediction setup with one critical modification: During pre-training, tokens corresponding to the retrieved factual values and the ending lookup token are excluded from the loss computation (Appendix A.3).

$$\mathcal{L}(\theta) = -\sum_{t=1}^{T} m_t \log p_\theta(x_t \mid x_{<t}), \quad m_t = \begin{cases} 0, & x_t \in \{\text{retrieved values}, \texttt{<|db\_end|>}^2\} \\ 1, & \text{otherwise} \end{cases} \quad (1)$$

This design **discourages LMLM from memorizing facts** that are offloaded to the external database. By interleaving lookup calls in the pre-training data together with the modified next-token prediction loss, it simplifies modeling factual content by decoupling knowledge learning into two steps: (1) the model learns to generate lookup calls, with arguments inferred from context; and (2) factual

---

[1] Specifically, we represent each fact as `<|db_start|>` entity `<|sep|>` relation `<|db_retrieve|>` value `<|db_end|>` in the pretraining data.

[2] We skip the loss on `<|db_end|>` since, after retrieval, the token can be inserted directly during generation and does not need to be learned.

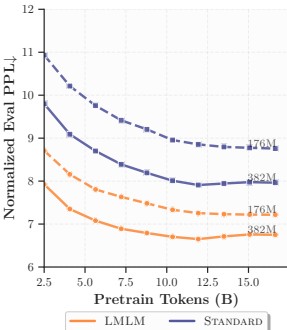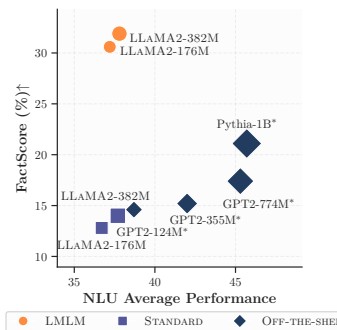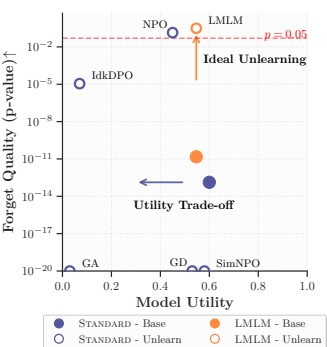

Figure 4: **Results overview.** *(Left)* LMLM achieves consistently lower perplexity during pre-training, indicating that offloading factual knowledge improves pre-training efficiency. *(Middle)* LMLM significantly improves factual precision over its STANDARD counterparts while maintaining NLU performance. *(Right)* On the TOFU machine unlearning benchmark, LMLM forgets targeted facts while preserving general model utility. Results shown for LMLM with a LLaMA backbone; * denotes off-the-shelf models.

knowledge is stored and retrieved explicitly from an external database rather than encoded in model parameters. This decoupling forms the core advantage of LMLM over the knowledge-dense counterparts. Intuitively, when the model can rely on accurate externally provided facts, it no longer needs to expend capacity learning complex, long-tail factual distributions. The resulting architecture can achieve better parameter efficiency by dedicating model capacity to reasoning and linguistic competencies. Empirically, this effect is reflected in faster convergence in pre-training and lower validation perplexity (Section 4.2).

**Inference** Similar to tool-augmented models (e.g., Toolformer (Schick et al., 2023)), LMLM generates text autoregressively until a special token triggers a database lookup, retrieves the corresponding value, and continues generation.

# 4 EXPERIMENTS

We evaluate LMLMs via validation perplexity (§ 4.2), machine unlearning (§ 4.3), as well as factual precision (§ 4.4), with summary of these results in Figure 4. We further discuss additional implications on LMLM in § 6.

## 4.1 EXPERIMENTAL SETUP

**Pretraining and Model Setup.** We pretrain on a high-quality Wikipedia corpus ($\sim$ 3B tokens) from the OLMo2 project[3] (Groeneveld et al., 2024), and evaluate perplexity on a held-out set of 1,000 samples ($\sim$ 245k tokens). We pre-train LMLM from scratch using GPT-2 and LLaMA2-style architectures with their standard tokenizers and vocabularies, extended by four special tokens for lookup calls. All models are trained for 8 epochs with a context length of 1,024 tokens, using mixed precision. Each model completes training within 8 H100-days. Details are in Appendix A.2.

**Database and Retrieval Setting.** We construct the database by annotating the entire pretraining corpus, resulting in 54.6M knowledge triplets. Retrieval uses fuzzy matching with cosine similarity over ALL-MINILM-L6-V2 embeddings (with a rejection threshold of 0.6).

**Baseline Comparisons.** We consider the following pre-training settings:

- LMLM *(Ours)*: Pre-trained on our annotated data with lookup calls, using the loss in Equation 1.
- STANDARD: Pre-trained on our data without lookup calls. All other settings are identical.
- OFF-THE-SHELF MODELS: Models with publicly available pre-training weights, including: OPENAI/GPT2-124M, OPENAI/GPT2-355M, OPENAI/GPT2-774M (Radford et al., 2019),

---

[3]https://huggingface.co/datasets/allenai/dolmino-mix-1124

PYTHIA-1B (Biderman et al., 2023), LLAMA2-7B (et al., 2023), and LLAMA3.1-8B (et al., 2024). OPENAI/GPT2-124M and OPENAI/GPT2-355M are comparable in size to our LMLM models. These models are marked with an asterisk (*) in the results tables.

## 4.2 LEARNING TO LOOKUP FACTS IS EASIER THAN MEMORIZATION

**Evaluation Setup.** We first evaluate our models using language modeling perplexity on a held-out Wikipedia validation set. As LMLM introduces lookup calls, its perplexity is not directly comparable to that of STANDARD. To ensure a fair comparison, we report three variants of perplexity:

- *Static (Oracle)*: Assumes perfect lookup behavior, where LMLM always generates correct lookup calls and retrieves the correct values. This provides an optimistic lower bound. Perplexity is computed over all tokens excluding the lookup calls.
- *Dynamic*: Reflects actual model behavior during inference. Lookup calls are generated and executed in real time, capturing failures due to incorrect queries or failed retrievals. Perplexity is again calculated over all tokens except the lookup calls.
- *Normalized*: Measures the combined likelihood of generating the correct queries and subsequent text. Perplexity is computed over all tokens except for the retrieved values, and normalized by the number of tokens in the original unannotated text. See Appendix A.3 for formal definitions.

**Perplexity Results.** Figure 5 reports validation perplexities for both LMLM and STANDARD models.

If LMLM were simply benefiting from retrievals without learning to query properly, we would expect both *Dynamic* and *Normalized* perplexity to remain high. Improvements across all variants indicate that LMLM is learning both to query and to generate more effectively.

We observe that LMLM consistently achieves lower perplexity than STANDARD across all model sizes and perplexity variants. In particular, LMLM

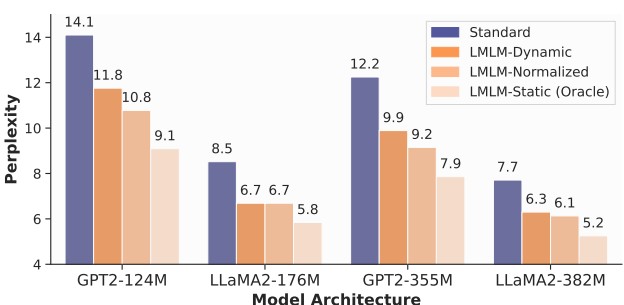

Figure 5: **Validation perplexity** comparison between STANDARD and LMLM on three variants of perplexity. Lower perplexity indicates better performance.

achieves an average perplexity reduction of 1.98 points under the *Dynamic* setting, demonstrating its effectiveness even with imperfect lookup calls. The *Normalized* variant highlights that LMLM assigns higher likelihoods to both lookup queries and grounded text, indicating improved training efficiency. These results support a key insight: Learning to lookup specific facts is easier than memorizing them.

**Downstream NLU performance.** We compare LMLM and STANDARD models on five standard natural language understanding (NLU) tasks (Table 14). This evaluation serves as a sanity check to ensure that separating factual knowledge during pretraining does not come at the expense of general language understanding. We find that LMLM performs on par with STANDARD across all tasks, confirming that factual offloading preserves the model's general-purpose capabilities.

## 4.3 MACHINE UNLEARNING: LMLM SUPPORTS INSTANT FORGETTING

One natural benefit of decoupling knowledge from model parameters is that editing and unlearning are achievable through simple operations on the external memory, without compromising the model's general capabilities. To verify this, we extend our evaluation to a standard machine unlearning benchmark, TOFU (Maini et al., 2024).

**Evaluation Setup.** TOFU evaluates unlearning efficacy in a privacy-sensitive setting, where the goal is to selectively forget a targeted subset of information (the *Forget Set*) while preserving performance on the *Retain Set* and maintaining general model capabilities. The objective is to produce an *unlearned model* that is statistically indistinguishable from a model trained solely on the Retain Set (referred to as the *retain model*). The benchmark consists of 200 synthetic author profiles, each containing 20 QA pairs. It evaluates two key aspects:

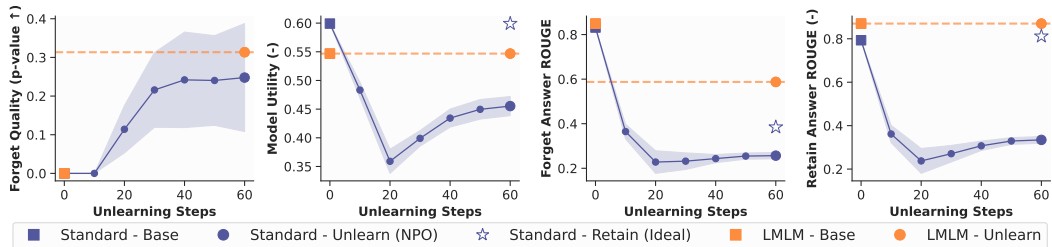

Figure 6: **Evaluation of Machine Unlearning.** We compare LMLM with NPO on the TOFU benchmark. Unlike prior methods, LMLM performs unlearning without any additional training. (a–b) Forget quality vs. utility trade-off. LMLM achieves ideal forgetting ($p$-value $> 0.05$) without sacrificing general utility. (c–d) LMLM retains knowledge outside the forget set, unlike other methods that degrade retain-set performance due to parameter entanglement. Y-axis labels use -- or ↑ to indicate retention or improvement. For metrics like Forget ROUGE, where lower isn't always better, we mark the retain model's performance as reference.

- *Model utility:* The average of three metrics—*ROUGE* (answer quality), *Answer Probability*, and *Truth Ratio* (likelihood assigned to the correct answer over distractors)—measured on the Retain Set, Real Author Set, and World Facts Set.
- *Forget quality:* The $p$-value of a statistical test comparing the unlearned model with the corresponding retain model to assess whether the targeted knowledge has been effectively removed.

We use LLAMA-3.2-1B-INSTRUCT as the base model and compare against NPO (Zhang et al., 2024b), a state-of-the-art unlearning method. For LMLM, we perform unlearning by simply removing entries in the database corresponding to the Forget Set. See Appendix B.2 for implementation details.

**Unlearning Results.** Figure 6 presents the results on the TOFU benchmark, where the Forget Set is 5% of the data. In Figure 6 (a-b), we show the forget quality and model utility throughout the unlearning process, where ideal performance is defined by effective forgetting without degrading model utility. LMLM achieves precisely this—effective forgetting with no loss in model utility—a direct benefit of decoupling factual knowledge from model parameters. Figure 6 (c–d) shows that LMLM preserves knowledge outside the Forget Set, whereas previous training-based methods tend to forget related information due to parameter entanglement. Importantly, forgetting is performed through simple database operations, without model updates or access to the Retain Set. In contrast, RL-based methods such as NPO, a state-of-the-art baseline on TOFU, incur a utility degradation, while other unlearning methods either fail to forget or exhibit catastrophic drops in model utility (See Figure 4, right). While LMLM incurs upfront costs for data annotation and model training, it provides substantial payoff in use cases where knowledge editing, removal, or compliance with data deletion requests are necessary.

### 4.4 EXTERNALIZING KNOWLEDGE IMPROVES FACTUAL PRECISION

We evaluate factual precision on long-form biography generation (FactScore) (Min et al., 2023) and short-form factual completion using T-REX (Petroni et al., 2021) and the long-tail subset of POPQA (Asai et al., 2023). FactScore measures the proportion of atomic facts supported by a trusted source, while T-REx and PopQA follow EM and Acc metrics from prior work (Schick et al., 2023; Asai et al., 2023).

As shown in Table 2, LMLM consistently improves factual preci-

Table 2: **Evaluations on factual precision.** Subscripts show the absolute difference from their respective STANDARD baselines.

| Model | Model Type | FactScore ↑ | T-REx EM ↑ | PopQA Acc ↑ |
|---|---|---|---|---|
| OPENAI/GPT2-124M* | - | 14.6 | 20.1 | 18.5 |
| GPT2-124M | STANDARD | 10.7 | 41.2 | 18.5 |
| | LMLM | **20.6**$_{+9.9}$ | **54.6**$_{+13.4}$ | **49.9**$_{+31.4}$ |
| LLAMA2-176M | STANDARD | 10.1 | 46.3 | 24.6 |
| | LMLM | **30.6**$_{+20.5}$ | **54.1**$_{+7.8}$ | **49.6**$_{+25.0}$ |
| OPENAI/GPT2-355M* | - | 15.2 | 28.4 | 19.1 |
| GPT2-355M | STANDARD | 14.4 | 44.9 | 21.4 |
| | LMLM | **23.9**$_{+9.5}$ | **58.7**$_{+13.8}$ | **52.0**$_{+30.6}$ |
| LLAMA2-382M | STANDARD | 14.0 | 52.0 | 22.7 |
| | LMLM | **31.9**$_{+17.9}$ | **58.1**$_{+6.1}$ | **50.8**$_{+28.1}$ |
| OPENAI/GPT2-774M* | - | 17.4 | 35.6 | 18.5 |
| PYTHIA-1B* | - | 21.1 | 47.8 | 19.5 |
| LLAMA2-7B* | - | 34.0 | 60.5 | 29.2 |
| LLAMA3.1-8B* | - | 40.3 | 67.3 | 29.4 |

sion over STANDARD, its knowledge-dense counterpart. On LLAMA2-382M, it gains $+17.9\%$ FactScore, $+6.1\%$ T-REx EM, and $+28.1\%$ PopQA Acc. Notably, LMLM approaches the performance of much larger models such as PYTHIA-1B and LLAMA2-7B, despite using far fewer parameters.

**How Does LMLM Compare to RAG?** Unlike Retrieval-Augmented Generation (RAG), which expands knowledge access at inference time, LMLM constrains factual memorization during pretraining by externalizing entity-level knowledge into a database. Since facts are not internalized in model weights, they can be instantly modified or removed through database operations alone (§ 4.3)—capabilities that are difficult to achieve with post-hoc retrieval. Current RAG systems still encode facts in parameters, so modifying specific knowledge requires model retraining or careful prompting to override internalized information (Xie et al., 2023; Hsia et al., 2024; Tan et al., 2024).

To contextualize their respective strengths, we compare against a controlled RAG baseline that retrieves the top-4 Wikipedia articles and prepends them to the standard LMs during generation. As shown in Table 3, LMLM achieves higher FactScore and PopQA accuracy at the current scale, indicat-

Table 3: Comparison of RAG vs. LMLM on factual precision. Results are shown for GPT2-355M.

| Model | FactScore ↑ | T-REx EM ↑ | PopQA Acc ↑ |
|---|---|---|---|
| OPENAI/GPT2-355M* | 15.2 | 28.4 | 19.1 |
| + RAG | 20.1 | 75.8 | 37.5 |
| GPT2-355M-LMLM | 23.9$_{+3.8}$ | 58.7$_{-17.1}$ | 52.0$_{+14.5}$ |

ing that it effectively queries its database for precise factual information. In contrast, RAG achieves stronger performance on T-REx, likely because the benchmark's queries are drawn verbatim from Wikipedia, making retrieval from the same corpus nearly perfect.

These results point to complementary strengths: RAG enriches pretrained models (including LMLM) with broader contextual information at inference, whereas LMLM restructures pretraining itself to provide fine-grained factual grounding that is directly editable and verifiable. A hybrid system could retrieve documents with RAG for contextual understanding while using LMLM for precise, verifiable entity lookups within those documents.

## 5 FURTHER ANALYSIS

**Does LMLM Still Memorize Facts in Its Parameters?** We provide preliminary evidence that LMLM reduces factual memorization through its training design. Using the TOFU synthetic trainset, we compare training objectives by tracking the loss on return value tokens—the factual answers intended to be retrieved rather than memorized.

Table 4: Disabling retrieval significantly reduces performance on both FACTSCORE and T-REx; see Table 10 for the full comparison.

| Model Type | FactScore ↑ | T-REx EM ↑ |
|---|---|---|
| STANDARD | 14.0 | 52.0 |
| LMLM (w/o database) | 12.8$_{-19.1}$ | 38.5$_{-19.6}$ |
| LMLM | 31.9 | 58.1 |

As shown in Figure 7, models trained with a standard SFT objective exhibit a rapid decrease in the loss on these tokens, suggesting memorization in the model's parameters. In contrast, LMLM, trained with the masked loss maintains a high loss throughout training, indicating that these facts are not stored internally. LMLM's successful application in the machine unlearning benchmark (§ 4.3) further supports this finding. Additionally, we observe a notable performance gap in Table 4, where factual precision drops substantially when the external database is disabled—forcing the model to rely on its internal parameters. These findings suggest that editing the database offers a direct way to control what the model knows and forgets.

**How Much Knowledge Should Be Externalized?** For our main experiments, we aim to exhaustively annotate all atomic entity-level knowledge. However, this may not be optimal. To explore the trade-off between storing knowledge in model parameters versus offloading it into the database, we devise a ranking criterion that quantifies the value of each lookup. Facts that require a lookup to achieve a high likelihood are difficult to learn—typically long-tail or highly specific knowledge—and should be externalized. On the other hand, externalizing facts that already have a high likelihood—typically common or easily inferred knowledge—provides limited upside and may harm general capabilities.

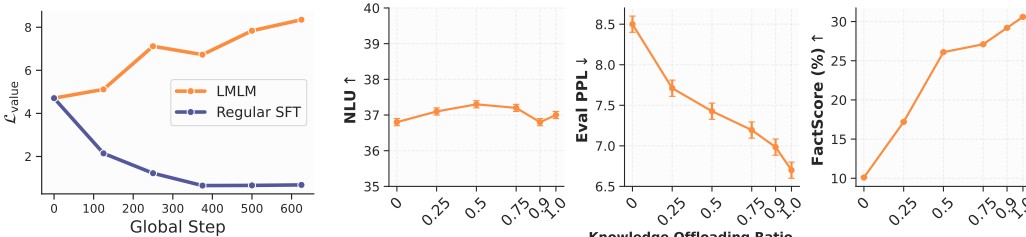

Figure 7: Training loss on return value tokens.

Figure 8: Effect of different offloading ratios; within our scope, more offloading is beneficial.

Concretely, we first train a LMLM model and a STANDARD model for one epoch on the data described in § 4. For each fact, we then measure the average loss difference on its factual value tokens. A large difference indicates that STANDARD LMs struggle to learn the fact (See Appendix B.3 for qualitative examples). Using this criterion, we pretrain LLMs with varying annotation ratios (e.g. For 90% knowledge offloading, we keep the top 90% knowledge with highest loss difference), interpolating between the two extremes of STANDARD (fully parametric) and LMLM (fully offloaded).

As shown in Figure 8, greater offloading consistently improves language modeling (lower perplexity) and factual precision, while preserving natural language understanding. This suggests that, for knowledge representable as triplets, increasing the offloading ratio is beneficial. Extending this spectrum to broader and more abstract forms of knowledge remains an open direction for future work.

## 6 DISCUSSION

**Toward Efficient Scaling via Knowledge Offloading.** We summarize our main findings in Figure 4. In particular, even smaller instances of LMLM match or exceed the performance of much larger off-the-shelf models. These results highlight the potential of LMLM to scale efficiently by offloading knowledge to an external database, thereby maintaining strong factual accuracy with fewer parameters. While our experiments are limited to modest scales due to computational constraints, the observed trends suggest that LMLMs offer a promising direction toward parameter-efficient language models that externalize factual storage. Such models may enable real-time, verifiable knowledge updates and open up new possibilities for deploying LMLM in resource-constrained or fact-sensitive environments.

**Beyond Entity-Level Knowledge Offloading.** By leveraging knowledge triplets, LMLM focuses on separating entity-level factual knowledge from linguistic competency. Current research on probing (Petroni et al., 2019; Hu et al., 2024), evaluating (Min et al., 2023; Wei et al., 2024), and editing (Meng et al., 2023a) internal knowledge in language models similarly concentrates on entity-level atomic facts. However, extending our method beyond entity-level knowledge remains a challenge. This difficulty arises in determining effective formats for separating knowledge beyond entity-focused triplets. For instance, using a simple QA function call format raises concerns about the potential for hallucinated facts during annotation.

Additionally, while LMLM attempts to minimize the memorization of factual knowledge, some knowledge still remains unremoved. More comprehensive benchmarks for probing internal knowledge in language models, as well as distinct benchmarks for disentangling knowledge and reasoning, remain underexplored.

**Limitations.** LMLM is a promising step towards separating factual knowledge from language models with many exciting future directions. Our current limitations include: (1) It does not guarantee perfect factuality during generation. Noise in the database and errors from fuzzy matching can introduce inaccuracies. However, such issues are easily traceable and verifiable for LMLM. (2) LMLM introduces additional tokens for lookup queries, which increases training and inference costs. (3) The current implementation focuses on entity-level factual knowledge, which captures only a subset of the broader knowledge spectrum. (4) Our experiments are limited to small models and datasets due to compute constraints. Although sufficient to show core benefits, scaling up may improve performance and support more complex tasks.

## 7 CONCLUSION

We introduce LMLMs, a new class of language models for externalizing knowledge alongside an integrated solution to achieve this. Our results demonstrate promising trends towards efficient use of model capacity and offloading facts onto an external database. LMLM represents an alternative way to store facts during pre-training, and has the potential to be integrated with other common approaches developed for LLMs, including retrieval-based methods, symbolic reasoning, as well as knowledge representation. Consequently, LMLM opens up new ways for future language models to leverage the benefits of external knowledge databases such as verifiable updates – fundamentally, it is much easier and more memory-efficient to learn how to look up facts rather than to remember them.

## ACKNOWLEDGMENTS

We thank Eric Enouen for valuable discussions and feedback. This research was supported by a gift to the LinkedIn–Cornell Bowers Strategic Partnership. SZ is supported by the Defense Advanced Research Projects Agency (DARPA) under Grant No. D24AP00259-00. CKB is supported by the National Science Foundation (NSF) through the NSF Research Traineeship (NRT) program under Grant No. 2345579. JPZ is supported by a grant from the Natural Sciences and Engineering Research Council of Canada (NSERC) (567916). DG is supported by Empire AI Postdoctoral Fellowship. This work is partially supported by Open Philanthropy; the National Science Foundation NSF under awards OAC-2311521, IIS-2107161, IIS-1724282, IIS-2505098 and HDR-2118310; NASA under award No. 20-OSTFL20-0053; the Cornell Center for Materials Research with funding from the NSF MRSEC program (DMR-1719875); DARPA; arXiv; Apple; AI-MI and NSF Award 2433348; and the New York Presbyterian Hospital. We gratefully acknowledge use of the research computing resources of the Empire AI Consortium, Inc, with support from the State of New York, the Simons Foundation, and the Secunda Family Foundation (Bloom et al., 2025). Any opinions, findings and conclusions or recommendations expressed in this material are those of the author(s) and do not necessarily reflect the views of the National Science Foundation or of NASA.

## ETHICS STATEMENT

This work complies with the ICLR Code of Ethics. We use only publicly available datasets and do not involve human subjects or private data. While our method is methodological in nature, we acknowledge general risks of bias and misuse inherent to language models.

## REPRODUCIBILITY STATEMENT

We open-source our code and models at `https://github.com/kilian-group/LMLM`. We provide full details of datasets, model configurations, and evaluation protocols in the main text and appendix to ensure that all results can be independently reproduced.

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

# A IMPLEMENTATION DETAILS

## A.1 KNOWLEDGE EXTRACTION DETAILS

**Model and Data.** We construct a high-quality seed dataset by sampling 1,000 passages from SQuAD-v2 (Rajpurkar et al., 2018) and 1,000 passages from Wikipedia. These passages are annotated by GPT-4o with structured factual triples and used to train the ANNOTATOR model. The remaining Wikipedia passages are used as the pre-training corpus for LMLM, with no overlap with the validation set.

Both the CORRECTOR and ANNOTATOR are based on LLAMA-3.1-8B-INSTRUCT, selected for its strong instruction-following capabilities. The CORRECTOR and ANNOTATOR use a maximum context length of 2048 tokens. All input sequences are truncated to 1024 tokens during LMLM pretraining for consistency.

GPT-4o and ANNOTATOR annotations follow the format `[dblookup('Entity', 'Relation') -> Value]`. These are converted to a token-based format for pretraining: `<|db_start|> Entity <|sep|> Relation <|db_value|> Value <|db_end|>`.

- The CORRECTOR is fine-tuned using LoRA (`r=32`, `alpha=16`) (Hu et al., 2021) for 2 epochs on 19k GPT-4o-annotated SQuAD-v2 passages, with a learning rate of $2 \times 10^{-4}$ and an effective batch size of 32. Sequence packing is enabled to improve training efficiency.

- The ANNOTATOR is tuned with instruction on the 2k annotated passages using LoRA (`r=32`, `alpha=16`), with a learning rate of $2 \times 10^{-4}$ and an effective batch size of 32. Training runs for 10 epochs with a maximum sequence length of 2048.

**Annotation Pipeline.** We adopt a three-stage pipeline to distill GPT-4o's structured annotations into a lightweight and scalable ANNOTATOR model:

Table 5: Prompt templates used for GPT-4o and ANNOTATOR annotation.

| Model | Prompt Template |
|---|---|
| GPT-4o | You are a knowledge base construction expert. Extract **entity-based factual knowledge** from a passage and annotate it using the format: `[dblookup('Entity', 'Relationship') -> Value]`. These annotations simulate a knowledge base query for factual generation. Place `dblookup` right after the entity and relationship appear, keeping the text flow natural. 
— 
**Entity-Based Factual Knowledge Principles:** 
 - Entities: Use full names for people, organizations, places, or works. 
 - Relationships: Use specific, reusable labels that define the connection clearly. 
 - Values: Keep them concise and factual. 
**Annotation Principles:** 
 1. Extract ALL Atomic Facts: Each annotation should capture a single verifiable fact. 
 2. Precise Annotations: Use correct and specific entity-relationship-value triples. 
 3. Ensure Reusability: Use standardized and reusable entity and relation names. 
 4. Contextual Positioning Rule: Place annotations only after both entity and relation appear. 
 5. Preserve Text and Maintain Flow: Do not alter or disrupt the original text. 
**Example Annotation:** 
*Input:* Beyoncé Giselle Knowles-Carter (born September 4, 1981) is an American singer, songwriter, record producer, and actress. 
*Output:* Beyoncé Giselle Knowles-Carter (born `[dblookup('Beyoncé Giselle Knowles-Carter', 'Birth Date') -> September 4, 1981]` September 4, 1981) is an `[dblookup('Beyoncé Giselle Knowles-Carter', 'Nationality') -> American]` American `[dblookup('Beyoncé Giselle Knowles-Carter', 'Occupation') -> singer, songwriter, record producer, actress]` singer, songwriter, record producer, and actress. |
| ANNOTATOR | Your task is to extract and annotate entity-based factual knowledge from the provided text. Identify and annotate specific entities, relationships, and values using the `dblookup` format: `[dblookup('Entity', 'Relationship') -> Value]` 
**Annotation Guidelines:** 
 - Inline Insertion: Insert `dblookup` before factual statements without altering the text. 
 - Atomic Facts: Each `dblookup` should capture one verifiable fact. 
 - Entities: Use full names for people, organizations, places, or works. 
 - Relationships: Use specific, reusable labels (avoid vague terms). 
 - Values: Keep them concise and factual. |

- **Stage 1: Seed Annotation.** GPT-4o is prompted to annotate input passages with structured factual triples (`entity, relation, value`). These are embedded directly into the text using the lookup call format. Prompt template is provided in Table 5.

- **Stage 2: Cleaning.** A warm-start CORRECTOR model is trained on the annotated data for 2 epochs without instruction prompts. Although underfit, it is effective at identifying noisy or ill-formed annotations. Specifically, we discard lookup calls where the token-level loss on the entity or relation is in the top 10% of the distribution.

- **Stage 3: Annotation.** We instruction-tune a new ANNOTATOR model on the cleaned dataset for 10 epochs. This model learns to detect when factual knowledge should be externalized and how to issue structured lookup queries. The trained model is then applied to the full pre-training corpus to generate large-scale factual supervision. Prompt template is provided in Table 5.

**Additional Notes.** We observe a bimodal loss distribution on entity and relation tokens in the GPT-4o-generated annotations. This is likely due to GPT-4o accessing future context during generation, which breaks the left-to-right constraint of autoregressive models. As a result, some annotations are not recoverable from preceding context alone.

The CORRECTOR helps filter out such cases—removing lookup calls that are (1) not inferable from prior context, (2) overly specific or inconsistent, or (3) syntactically malformed. This filtering improves the quality of supervision provided to the final ANNOTATOR model.

Ultimately, the ANNOTATOR learns to insert lookup calls only when they are contextually grounded and likely to enhance factual accuracy. This encourages retrieval-based reasoning and helps LMLM offload factual knowledge from its parameters into a structured database.

**Additional details of the evaluation and ablation study for the annotation pipeline.** We evaluated different annotator training configurations on a GPT-4 annotated seed evaluation set containing 100 SQuAD passages and 100 Wikipedia articles, which were cleaned to ensure reliable supervision signals. The ANNOTATOR was LoRA-finetuned for 10 epochs on a mixed dataset of 1k SQuAD and 1k Wikipedia examples with the intermediate filtering step. Two ablation baselines were compared: *w/o data mixing*, which trained only on 19k SQuAD while keeping total compute constant, and *w/o filtering*, which trained on the same mixed data but without the filtering stage.

As shown in Table 6, the mixed-data and filtering design achieves the lowest losses across both domains, confirming that both steps contribute meaningfully to annotation quality.

Table 6: Loss decomposition on the original corpus and the average loss over lookup calls for different annotator training configurations.

| Model | Wiki (orig.) | Wiki (lookup avg.) | SQuAD (orig.) | SQuAD (lookup avg.) |
|---|---|---|---|---|
| ANNOTATOR (ours) | **3.33e-3** | **7.85e-2** | **2.93e-3** | **1.22e-1** |
| *- w/o data mixing* | 4.21e-3 | 1.18e-1 | 3.67e-3 | 1.81e-1 |
| *- w/o filtering* | 3.38e-3 | 8.21e-2 | 3.31e-3 | 1.30e-1 |

## A.2 MODEL ARCHITECTURE AND TRAINING DETAILS

We pretrain LMLM from scratch using GPT-2 and LLaMA2-style decoder-only architectures. Each model uses its original tokenizer and vocabulary, extended with four special tokens reserved for lookup calls. This results in a vocabulary size of 50,261 for GPT-2 models and 32,004 for LLaMA2 variants. Full architecture specifications, including hidden size, depth, and parameter counts, are shown in Table 7.

All models are trained for 8 epochs with a context length of 1,024 tokens using mixed-precision training. For LLAMA2-176M and LLAMA2-382M, we use a batch size of 256 and train for 105k steps, totaling approximately 8 H100-days. Training is performed using Hugging Face Accelerate in `bf16` precision. Hyperparameters such as learning rate, scheduler, and warmup steps are detailed in Table 8.

Table 7: Model architecture, vocabulary size (including 4 special tokens), and parameter counts. We report both total and non-embedding parameter counts.

| Model | Hidden Size | #Layers | #Heads | Vocab Size | Params (Total / Non-Embed) |
|---|---|---|---|---|---|
| GPT2-124M | 768 | 12 | 12 | 50,261 | 124.4M / 85.5M |
| LLaMA2-176M | 512 | 8 | 8 | 32,004 | 176.4M / 160.0M |
| GPT2-355M | 1024 | 24 | 16 | 50,261 | 354.8M / 303.4M |
| LLaMA2-382M | 768 | 12 | 12 | 32,004 | 381.8M / 357.3M |

Table 8: Training hyperparameters. LLaMA2-176M, LLaMA2-382Mare initialized in $float32$ and trained with mixed precision ($bf16$) using Hugging Face Accelerate.

| Model | Batch Size | Total Steps | LR | Scheduler | Warmup | Precision |
|---|---|---|---|---|---|---|
| GPT2-124M,GPT2-355M | 320 | 66k | 5.0e-4 | – | – | $bf16$ (mixed) |
| LLaMA2-176M, LLaMA2-382M | 256 | 105k | 5.0e-4 | cosine | 2000 | $bf16$ (mixed) |

## A.3 FORMALIZATION OF TRAINING AND EVALUATION OBJECTIVES

We denote an autoregressive language model by $p_\theta$, which defines a probability distribution over a sequence of tokens $x = (x_1, \ldots, x_T)$ as:

$$p_\theta(x) = \prod_{t=1}^{T} p_\theta(x_t \mid x_{<t}).$$

Each token $x_t$ belongs to one of the following categories:

- $\mathcal{T}_{\text{org}}$: original text tokens from the raw corpus;
- $\mathcal{T}_{\text{train}}$: Return values and `<|db_end|>` are excluded from training loss objective.
- $\mathcal{T}_{\text{e}}, \mathcal{T}_{\text{r}}$: tokens representing entities and relation arguments within database lookup calls;
- $\mathcal{T}_{\text{v}}$: tokens corresponding to retrieved factual values (i.e., return values);
- $\mathcal{T}_{\text{db}}$: special tokens used to mark database lookup segments, including:
  - `<|db_start|>`: begins a lookup call;
  - `<|sep|>`: separates entity and relation in the query;
  - `<|db_retrieve|>`: signals the insertion point for the returned value;
  - `<|db_end|>`: marks the end of the lookup block.

Here is an example using background color to highlight different token categories:

$\mathcal{T}_{\text{org}}$:
```
Napoleon was born on  <|db_start|> Napoleon <|sep|> Birth_Date
<|db_retrieve|> August 15, 1769 <|db_end|>  August 15, 1769.
```

$\mathcal{T}_{\text{train}}$:
```
Napoleon was born on <|db_start|> Napoleon <|sep|> Birth_Date <|db_retrieve|>
August 15, 1769 <|db_end|>  August 15, 1769.
```

**Training Loss.** The training objective excludes supervision over return values and the closing marker `<|db_end|>` to prevent memorization of factual knowledge:

$$\mathcal{L}(\theta) = - \sum_{t \in \mathcal{T}_{\text{train}}} \log p_\theta(x_t \mid x_{<t}), \quad \text{where } \mathcal{T}_{\text{train}} = \{t \mid x_t \notin \mathcal{T}_{\text{v}} \cup \{\texttt{<|db\_end|>}\}\}. \quad (2)$$

**Evaluation Metrics.** We report both perplexity and negative log-likelihood (NLL), computed over different token subsets depending on the evaluation setting:

- **Static & Dynamic Perplexity:** Tokens corresponding to lookup calls and return values are excluded:

$$\text{PPL}_{\text{static/dynamic}} = \exp\left(-\frac{1}{|\mathcal{T}_{\text{org}}|}\sum_{t \in \mathcal{T}_{\text{org}}} \log p_\theta(x_t \mid x_{<t})\right).$$

- **Normalized Perplexity:** This metric fairly compares generation likelihood by excluding retrieved factual values from the loss, but normalizing by the length of the original (fully reconstructed) text. Specifically:

$$\text{PPL}_{\text{norm}} = \exp\left(-\frac{1}{|\mathcal{T}_{\text{org}}|}\sum_{t \in \mathcal{T}_{\text{train}}} \log p_\theta(x_t \mid x_{<t})\right)$$

- **Negative Log-Likelihood (NLL):** Matches the training loss computation:

$$\text{NLL}(x) = -\sum_{t \in \mathcal{T}_{\text{train}}} \log p_\theta(x_t \mid x_{<t}).$$

## A.4 DATABASE AND RETRIEVAL SETTING

We build our database by annotating the pre-training data, obtaining 54.6M knowledge triplets consisting of 9.5M unique entities, 8.5M relationships and 16.2M unique values. For retrieval, we employ a fuzzy matching mechanism based on the cosine similarity of sentence embeddings from ALL-MINILM-L6-V2[4]. As shown in Figure 9, specifically, given a lookup call, we compute its embedding and compare it with the embeddings of stored triplets in our database. If the highest similarity score is below a threshold of 0.6, we return *unknown* to indicate that no sufficiently similar match was found. Alternatively, we implement a prefix-tree constrained generation, ensuring that lookup calls remain covered by the structured knowledge representations. See detailed discussion in Appendix D.3.

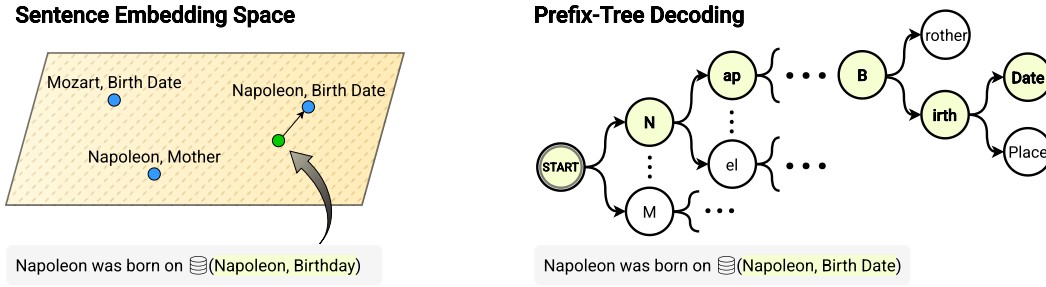

Figure 9: Unconstrained vs. Constrained Query Generation.

---

[4]https://huggingface.co/sentence-transformers/all-MiniLM-L6-v2

## B   EXPERIMENTAL SETUP

### B.1   EVALUATION BENCHMARKS

**Perplexity.**   We evaluate language modeling perplexity on a held-out Wikipedia test set consisting of 1,000 passages (~245k tokens). We use the same tokenizer as the original model (either GPT2 or LLaMA2) and apply Hugging Face's Trainer with sequence packing enabled. For LMLM, reference completions are annotated using ANNOTATOR. In the *dynamic* setting, the model generates its own lookup arguments, but we force lookup calls to occur at the same positions as in the reference. The exact formulation of perplexity is provided in Appendix A.3.

**FactScore.**   We evaluate factual precision using FACTSCORE (Min et al., 2023), a benchmark for open-ended biography generation. Given a generated text, FactScore extracts a set of atomic facts and computes the proportion that is supported by a trusted knowledge source. We use the first 100 biography queries provided in the benchmark. All models generate outputs using greedy decoding (maximum length = 256 tokens; repetition penalty = 1.2). Factuality is validated using retrieval-augmented prompting with ChatGPT, following the official evaluation protocol.[5]

For LMLM, which is not instruction-tuned, we use a fixed prompt template to elicit biography completions: "`Tell me a bio of <name>.  <name> is`" This prompt is applied consistently across all samples. To encourage structured queries during generation, we apply a logit bias to four special tokens in the vocabulary: `<|db_start|>`, `<|sep|>`, `<|db_retrieve|>`, and `<|db_end|>`, with respective bias values of 5.0, 2.0, 2.0, and 2.0. Retrieval is performed using fuzzy matching with a cosine similarity threshold of 0.6. If no relevant triplet is found, the model continues generation using the plain text string `unknown` as a fallback. This behavior is untrained and left for future work to improve robustness to retrieval failures.

**T-REx.**   We adapt the T-REx subset from LAMA (Petroni et al., 2019) for autoregressive models by filtering examples in which the masked entity does not appear in the final position. This yields 11,615 left-to-right compatible examples, following Schick et al. (2023). Each input consists of a factual statement, such as: "`Jaideep Sahni (born 1968) is an Indian [MASK]`" The model is expected to complete the statement with a single token (e.g., `actor`). We evaluate using two metrics: *Exact Match*, which checks whether the reference answer appears among the first five generated content words, and *Precision@1*, which checks whether the first generated content token matches the reference.

All models generate outputs using greedy decoding, and both metrics are computed after post-processing to remove lookup calls. For LMLM, we enforce a database lookup call at the masked position using fuzzy matching with a similarity threshold of 0.6. If no match is found, the model continues with standard decoding without triggering a structured lookup call, as the target fact may belong to common knowledge not covered by the database.

**PopQA.**   We evaluate on the long-tail subset of PopQA, which contains 1,399 queries about rare entities (fewer than 100 monthly Wikipedia page views), following Asai et al. (2023). Performance is measured by *Exact Match* (EM), i.e., whether the gold answer appears in the model output. To enable fair comparison with small pre-trained models that are not instruction-tuned, we convert the QA format into a knowledge-completion task by prompting GPT-4 to rewrite each query. This reduces dependence on instruction-following ability and allows answers to be appended directly. All results reported in the paper use this rewritten format. For example: Original: `What is Ufa the capital of?` Rewritten: `What is Ufa the capital of?  Ufa is the capital of.`

### B.2   MACHINE UNLEARNING SETTING

**TOFU.**   The TOFU benchmark (Maini et al., 2024) evaluates unlearning efficacy in privacy-sensitive scenarios, aiming to selectively remove a specific subset of information (the *Forget Set*) from a model (the *Full Model*) while maintaining performance on retained information (the *Retain Set*) and general

---

[5]`https://github.com/shmsw25/FActScore`

model capabilities. The benchmark's primary goal is to ensure the resulting *Unlearned Model* is statistically indistinguishable from a model trained exclusively on the Retain Set (the *Retain Model*).

The TOFU benchmark comprises 200 synthetic author profiles, each associated with 20 QA pairs. Performance is assessed using two primary metrics:

- *Model utility:* An average of three metrics—*ROUGE* (answer quality), *Answer Probability* (the likelihood of correct answers), and *Truth Ratio* (the likelihood assigned to correct answers over distractors)—evaluated on the Retain Set, Real Author Set, and World Facts Set.
- *Forget quality:* The $p$-value from a statistical test comparing the *unlearned model* to the corresponding *retain model*, quantifying the effectiveness of knowledge removal.

Our implementation directly builds upon the official TOFU repository[6].

We use LLAMA-3.2-1B-INSTRUCT as the base model and compare against NPO (Zhang et al., 2024b), a state-of-the-art method for unlearning. We test on forget 5% setting. For LMLM, unlearning is implemented simply by removing relevant entries corresponding to the Forget Set from the external database.

We adopt LLAMA-3.2-1B-INSTRUCT as the backbone for two reasons: (1) TOFU is designed for instruction models, and the official benchmark reports all baselines using instruction-tuned backbones; and (2) TOFU is synthetic, so using an instruction model that has never seen the underlying knowledge provides a fair initialization to show that LMLM prevents knowledge internalization by design. Additional results on pretrained LMLM is in Appendix C.4.

**NPO baseline.** Negative Preference Optimization (NPO) (Zhang et al., 2024b) is a state-of-the-art method for selective knowledge removal, especially effective for unlearning large portions (50%-90%) of training data. Unlike traditional gradient-based unlearning methods, which often degrade a model's general performance and struggle to forget only 10% of training data, NPO explicitly guides the model away from undesired (negative) samples, maintaining stable training dynamics and preventing catastrophic performance degradation. Further details can be found in the original paper (Zhang et al., 2024b). We follow the official TOFU implementation, running unlearning fine-tuning three times with different random seeds and reporting the mean and variance in Figure 6.

**LMLM Implementation of TOFU Evaluation.** To evaluate unlearning effectiveness, we assume the synthetic knowledge used in TOFU is fully represented in our external database. We annotate the complete TOFU dataset (4k synthetic QA pairs) using GPT-4o, subsequently building our database from these annotations.

Since LMLM is applied to pre-existing models, we introduce an additional step to ensure the model can utilize the lookup mechanism effectively. Specifically, we first perform a warm-up training stage on annotated Wikipedia data, followed by fine-tuning on the annotated TOFU training set, using the same hyperparameters as the baseline models (see Table 9).

For ROUGE evaluations, generated answers are post-processed to remove structured lookup tokens before computing the scores. For likelihood-based metrics (Answer Probability and Truth Ratio), we evaluate the model's probabilities only on the annotated training input segments ($\mathcal{T}$_train, as defined in Appendix A.3). Thus, ROUGE scores remain directly comparable across methods, but likelihood-based metrics are not directly comparable due to differences between raw and annotated reference answers.

**Interpreting TOFU Results.** When interpreting TOFU results, the key indicator of successful unlearning is how closely the *unlearned model*'s performance matches that of the *retain model*. Ideally, the two models should be indistinguishable, indicated by a forget quality $p$-value above 0.05. In practice, this means the *unlearned model* should achieve forget quality above 0.05 while maintaining consistent model utility, and its ROUGE scores on both the Forget Set and Retain Set should closely resemble those of the *retain model*. Importantly, lower ROUGE scores on the Forget Set do not necessarily imply better unlearning, as these reductions could also result from general performance degradation.

---

[6]https://github.com/locuslab/open-unlearning

## B.3    Experimental Setting of Further Analysis

**Selective Knowledge Offloading.**    We conduct a preliminary study on the trade-off between storing knowledge in model parameters and offloading it into the database. To enable *selective externalization*, we revert only a subset of annotations based on model learning difficulty.

Specifically, we first train a LmLm model with the original annotated data described in Sec. 4 and a Standard model with the unannotated data for one epoch. We measure the difference of the language modeling loss for the 5 tokens following each lookup for the LmLm and the corresponding tokens in the clean document for the Standard model. A larger gap suggests that Standard struggles to memorize the fact—typically long-tail or highly specific knowledge—making it a strong candidate for externalization. A smaller gap indicates that (i) the fact can be easily memorized by Standard, (ii) contextual hints make it trivial, or (iii) database lookups in LmLm are noisy or less useful. We calculate this loss difference across the full Wikipedia dataset, and include the distribution (Figure 12) and random qualitative examples from each quantile bucket for illustration (Table 19).

Based on this criterion, we then pretrain LLMs with varying *knowledge offloading ratios* $[0\%, 25\%, 50\%, 75\%, 90\%, 100\%]$. Here, $0\%$ corresponds to Standard (fully parametric) and $100\%$ to LmLm (fully externalized). Offloading is applied progressively, starting with triplets that show the largest loss differences. The corresponding loss thresholds are: 3.51 for 10% offloading, 2.24 for 25%, 1.25 for 50%, 0.64 for 75%, and 0.13 for 90%.

## B.4    Experimental Settings by Figure

We detail the experimental configurations corresponding to each figure in the main paper:

- **Figure 4**: (Left) We eval on a held-out wikipedia validaition set of 100 passages (~21k tokens) every 1000 steps during pretraining. (Middle) The detailed results for FactScore and NLU are in Table 2 and  14. (Right) The backbone model is LLaMA-3.2-1B-Instruct. We implement LmLm evaluation based on TOFU official repo It is forget 5% setting. The detailed results for baselines methods are copied from TOFU repo.

- **Figure 6**: For NPO baselines, we follow the official TOFU implementation, running unlearning fine-tuning five times with different random seeds (0, 42, 420, 69, 4497) and reporting the mean and variance throughout training. The ideal performance of the *retain model* is indicated with a marker. For LmLm, which does not require training to unlearn, we show only the pre- and post-unlearning results. Details on how to interpret the results are in Appendix B.2.

- **Figure 7**: We use the TOFU synthetic training set to compare training objectives by tracking the loss on return value tokens—the factual spans intended to be retrieved via lookup. Models are evaluated every 125 steps during training.

Table 9: Training hyperparameters used in different experimental settings. (Full set: 4k QA pairs, Retain set: 3.8k QA pairs)

| Setting | learning_rate | warmup_steps | num_train_epochs | batch_size | dataset |
|---|---|---|---|---|---|
| Finetune (Standard) | 1e-5 | 0.2 | 5 | 32 | TOFU trainset |
| Warmup (LmLm) | 5e-5 | 0.25 | 1 | 64 | Annotated Wikipedia (9.8k chunk) |
| Finetune (LmLm) | 5e-5 | 0.2 | 5 | 32 | Annotated TOFU trainset |

# C DETAILED RESULTS

## C.1 ADDITIONAL NOTES ON RAG.

We include a standard retrieval-augmented generation (RAG) baseline as a point of reference, following Lewis et al. (2021). The retriever uses BM25 (via the `BM25Retriever` from FlashRAG[7]) to select the top-4 relevant chunks from English Wikipedia, segmented into 100-word passages. These passages are prepended to the model input using the prompt format: "`Answer the question or complete the prompt based on the given document. The following are given documents: [RETRIEVED_DOCUMENTS] \n [USER_QUERY]`" (see Table 23). Retrieval is performed at inference time only, without fine-tuning. We evaluate RAG using OPENAI/GPT2-355M to match the scale of LMLM.

As an additional note on Table 3, RAG performs reasonably well on FactScore, and we expect further improvements with larger, instruction-tuned models, as smaller models may struggle to effectively use retrieved context. RAG also achieves high scores on T-REx, though this may reflect the benchmark's overlap with Wikipedia, where many completions are retrieved verbatim. We include RAG results to provide a broader empirical context.

While both RAG and LMLM access external knowledge, they differ fundamentally. RAG retrieves unstructured text from Wikipedia and relies on the model to extract relevant content. In contrast, LMLM learns to issue explicit, structured lookups only when needed, interleaving retrieval with generation at the entity level. This enables more precise and easily verifiable access to factual knowledge. The two approaches are complementary—RAG could be applied on top of LMLM for potential additional gains.

## C.2 FACTUAL DEGRADATION WHEN FORCING INTERNAL RECALL.

Table 10 shows the full results when database access is disabled, forcing models to rely solely on internal parameters. Across all variants, we observe a consistent drop in factual precision, often below the STANDARD baseline. This degradation supports our main claim: LMLM does not memorize factual answers but retrieves them externally. These results highlight that what the model knows and forgets is determined by the database, enabling precise and direct control through simple edits.

Table 10: Impact of database access on factual precision. Disabling access leads to performance drops in both FACTSCORE and T-REx, confirming that LMLM relies on retrieval from external database rather than memorization.

| Model | Model Type | Database | Metrics | |
| --- | --- | --- | --- | --- |
| | | | FActScore (%) ↑ | T-REx Exact Match (%) ↑ |
| GPT2-124M | STANDARD | - | 10.7 | 41.2 |
| | LMLM | × | $14.9_{-5.7}$ | $32.0_{-22.6}$ |
| | LMLM | ✓ | 20.6 | 54.6 |
| LLAMA2-176M | STANDARD | - | 10.1 | 46.3 |
| | LMLM | × | $11.3_{-19.3}$ | $34.9_{-19.2}$ |
| | LMLM | ✓ | 30.6 | 54.1 |
| GPT2-355M | STANDARD | - | 14.4 | 44.9 |
| | LMLM | × | $10.4_{-13.5}$ | $36.4_{-22.3}$ |
| | LMLM | ✓ | 23.9 | 58.7 |
| LLAMA2-382M | STANDARD | - | 14.0 | 52.0 |
| | LMLM | × | $12.8_{-19.1}$ | $38.5_{-19.6}$ |
| | LMLM | ✓ | 31.9 | 58.1 |

## C.3 ADDITIONAL LOSS CURVES.

Figure 10 shows the full training loss curves on different token types in the TOFU synthetic trainset. While the main paper focuses on return value tokens, we include loss on $\mathcal{T}_{\text{org}}$, entity, and relationship

---

[7]https://github.com/RUC-NLPIR/FlashRAG/blob/main/docs/original_docs/baseline_details.md#global-setting

tokens here for completeness. We observe no significant difference between training objectives on these tokens.

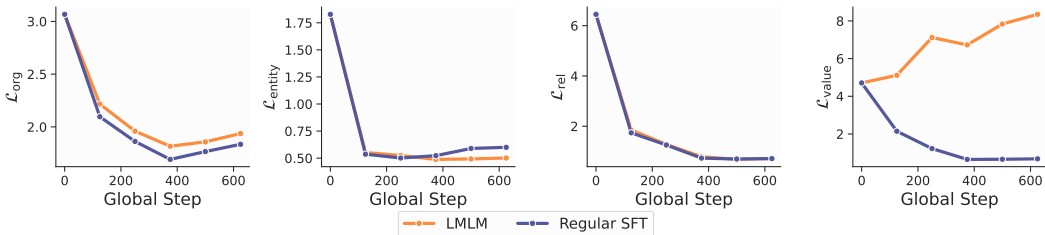

Figure 10: Training loss on $\mathcal{T}_{\text{org}}$ tokens, entity tokens, relationship tokens, and return value tokens.

## C.4 ADDITIONAL TOFU RESULTS ON PRETRAINED LMLM.

Evaluating our *pretrained* LMLM models on TOFU is also valuable. We now include results using LLAMA2-382M, which was trained from scratch under the LMLM paradigm. As shown below, both LMLM variants achieve effective unlearning (p-value $> 0.05$) without degrading model utility, consistent with the results in Section 4.3.

Table 11: TOFU unlearning results. "Model utility" measures preservation of general model behavior; "forget quality" is measured via a two-sample test (higher p-value indicates better unlearning). Baseline numbers use Llama-3.2-1B-Instruct from the official TOFU repository.

| Method | Model Utility ($-$) | Forget Quality (p-value ↑) |
|---|---|---|
| Finetuned | 0.60 | 1.33e-13 |
| Retain (Ideal) | 0.60 | 1.00 |
| GradAscent | 0.00 | 1.94e-119 |
| GradDiff | 0.53 | 1.94e-119 |
| IdkDPO | 0.07 | 1.12e-05 |
| NPO | 0.45 | 0.14 |
| SimNPO | 0.58 | 5.01e-100 |
| RMU | 0.58 | 4.87e-10 |
| LMLM (LLAMA-3.2-1B-INSTRUCT) | 0.55 | 0.31 |
| LMLM (LLAMA2-382M) | 0.43 | 0.70 |

## C.5 COMPARISON WITH MEMORY-BASED RELATED WORK.

Table 12: Comparison of LMLM against prior memory-based work on EM (%). All baseline numbers are reported from their original papers. "Params" refers to non-embedding model parameters. "External Memory" summarizes the format or corpus scale used at inference.

| Model | Params | External Memory | T-REx EM (%) |
|---|---|---|---|
| CoLAKE | 0.11B | ✓ 5M Wikidata Triples | 28.8 |
| K-Adapter | 0.34B | ✓ 42M Adapter Parameters | 29.1 |
| BERT-KNN | 0.34B | ✓ 900M Instance Datastore | 38.7 |
| EaE | 0.11B | ✓ 1M Entity Memory | 38.6 |
| FILM | 0.11B | ✓ 1.7M Wikidata Triples | 44.2 |
| Toolformer | 6.0B | ✓ API Access to Wikipedia | 53.5 |
| **LMLM** | 0.16B | ✓ 54.6M Triples | 54.1 |
| **LMLM** | 0.36B | ✓ 54.6M Triples | **58.1** |

We also compare LMLM with prior memory-augmented language models (Table 13). Several approaches externalize factual knowledge by attaching explicit memory modules. Fact-Injected Language Model (FILM) (Verga et al., 2021) augments LMs with a neuro-symbolic memory of entity–fact pairs. CoLAKE (Sun et al., 2020) jointly pretrains language and knowledge representations through a modified Transformer encoder. BERT-kNN (Kassner & Schütze, 2020) and EaE (Févry et al., 2020) use kNN-style retrieval or entity-specific memories to handle rare entities. K-Adapter (Wang et al., 2020) injects structured knowledge into PLMs via adapter modules. KBLaM (Wang et al., 2024)

Table 13: Comprehensive comparison of related methods. $\checkmark$ = supported, $\times$ = not supported, and $\sim$ = partial support (e.g., edit without unlearning, or not explicitly evaluated in prior work).

| Model | Architecture | Knowledge Storage | | | Performance | | | Goal |
|---|---|---|---|---|---|---|---|---|
| | | Internal | External | Integration | PPL↓ | Factuality↑ | Unlearning | |
| *Pretraining* | | | | | | | | |
| REALM (Guu et al., 2020) | Enc-only | $\checkmark$ | Docs | Prompt prepend | $\checkmark$ | $\checkmark$ | $\times$ | Joint retriever + LM |
| RETRO (Borgeaud et al., 2022) | Dec-only | $\checkmark$ | Docs | Cross-attn. to retrieved chunks | $\checkmark$ | $\checkmark$ | $\times$ | Scale with retrieval |
| Atlas (Izacard et al., 2022) | Enc–dec | | Docs | FiD (passages) | $\checkmark$ | $\checkmark$ | $\times$ | Few-shot generalization |
| Memorizing Transformer (Wu et al., 2022) | Dec-only | $\checkmark$(params+cache) | – | kNN attention (cache) | $\checkmark$ | $\times$ | $\times$ | Long-term memory |
| MemSinks (Ghosal et al., 2025) | Dec-only | $\checkmark$(sink neurons) | – | Memorization isolation | $\checkmark$ | $\times$ | $\checkmark$ | Localize memorization |
| Memory[3] (Yang et al., 2024) | Dec-only | $\checkmark$(partial) | Sparse KV | Self-attention | $\checkmark$ | $\checkmark$ | $\times$ | Reduce parametric memorization |
| **LMLM (ours)** | Dec-only | $\checkmark$(partial) | KB | Explicit lookup | $\checkmark$ | $\checkmark$ | $\checkmark$ | **Decouple facts from weights** |
| *Post-training* | | | | | | | | |
| K-Adapter (Wang et al., 2020) | Enc + adapters | $\checkmark$(params+adapt.) | – | Adapter layers | $\times$ | $\checkmark$ | $\sim$ | Inject KB via adapters |
| RAG (Lewis et al., 2021) | Enc-only | $\checkmark$ | Docs | Retrieved docs in prompt | $\times$ | $\checkmark$ | $\times$ | Improve factuality |
| Toolformer (Schick et al., 2023) | Dec-only | $\checkmark$ | APIs | Tool calls | $\times$ | $\checkmark$ | $\times$ | Extend via tools |
| MEMIT (Meng et al., 2023b) | Dec-only | $\checkmark$ | – | Weight editing | $\times$ | $\times$ | $\checkmark$ | Direct fact edit |
| *Inference* | | | | | | | | |
| kNN-LM (Shi et al., 2022) | Dec-only | $\checkmark$ | DS | kNN search + prob. interp. | $\checkmark$ | $\checkmark$ | $\times$ | Neighbor interpolation |
| RAG-variant | Dec-only | $\checkmark$ | Docs | Retrieved docs in prompt | $\times$ | $\checkmark$ | $\times$ | LLM + retrieval |
| *Memory-based Models* | | | | | | | | |
| CoLAKE (Sun et al., 2020) | Enc-only | $\checkmark$ | KG | Joint MLM over KG | $\checkmark$ | $\checkmark$ | $\times$ | Joint language + knowledge repr. |
| BERT-kNN (Kassner & Schütze, 2020) | Enc-only | $\checkmark$ | Docs | kNN search | $\times$ | $\checkmark$ | $\times$ | QA for long-tail facts via retrieval |
| EaE (Févry et al., 2020) | Enc-only | $\checkmark$ | Entity Mem | Entity Mem + prob. interp. | $\times$ | $\checkmark$ | $\times$ | Entity-specific memories |
| FILM (Verga et al., 2021) | Enc-only | $\checkmark$ | KB | Entity-fact Mem + prob. interp. | $\times$ | $\checkmark$ | $\sim$ | Neuro-symbolic KB with fact injection |
| KBLaM (Wang et al., 2024) | Dec-only | $\checkmark$ | Continuous KV | Modified rectangular attention | $\times$ | $\checkmark$ | $\times$ | Integrate KB tokens |
| MemLLM (Modarressi et al., 2025) | Dec-only | $\checkmark$ | Extracted triplets | Read-write calls | $\checkmark$ | $\checkmark$ | $\sim$ | FT to use read-and-write memory |
| MLP-Memory (Wei et al., 2025) | Dec-only | $\checkmark$ | MLP-based | MLP imitates kNN retrieval | $\checkmark$ | $\checkmark$ | $\times$ | High-throughput memory |

augments knowledge in the form of continuous key-value vectors as knowledge tokens and integrates them into pre-trained LLMs with a modified rectangular attention structure. MemLLM (Modarressi et al., 2025) proposes an explicit read-write memory, enabling LLM to extract knowledge into triplets, stores and later recalls knowledge for specific tasks. Concurrently, MLP-Memory (Wei et al., 2025) distills a kNN retriever into a lightweight MLP without accessing documents, which improves factuality and inference throughput.

While these methods improve factuality by coupling models with external memories, they generally do not address controllable unlearning, which distinguishes LMLM from this line of work.

Table 12 shows that LMLM achieves consistently higher precision (Exact Match) on T-REx compared to prior memory-augmented and tool-augmented models, while using fewer parameters and without depending on large external tool APIs. These results highlight both the modeling advantages of our approach and its empirical gains over existing memory-based baselines.

Additionally, Memory[3] (Yang et al., 2024) pretrains models to write and read sparse attention key-values as explicit memories, reducing the burden of model parameters to memorize specific knowledge. Memory[3] shares the similar motivation with LMLM that both pretrain models from scratch to access external memory to reduce parametric memorization. However, LMLM benefits from an external natural-language knowledge database, which is easy to update, verify and unlearn. In addition, LMLM offers a conceptually simple training recipe that can be integrated by finetuning existing LMs on knowledge-intensive domains.

# D FURTHER ANALYSIS

## D.1 DOES LMLM AFFECT LANGUAGE UNDERSTANDING?

Beyond the promising results, it is important to verify that our approach does not compromise the general capabilities of pretrained language models. To assess this, we follow Penedo et al. (2024) and evaluate on a set of "high-signal" Natural Language Understanding (NLU) benchmarks (as shown in Table 14). Given our focus on smaller models, we exclude benchmarks where both GPT-2 and similarly sized models fail to rise above the noise floor (Du et al., 2025). This leaves us with the following benchmarks: *Commonsense QA* (Talmor et al., 2019), *HellaSwag* (Zellers et al., 2019), *PIQA* (Bisk et al., 2019), *SIQA* (Sap et al., 2019), and *ARC Easy* (Clark et al., 2018). Implementation details can be found in lighteval[8].

It is important to mention that STANDARD and LMLM are pretrained solely on the Wikipedia dataset, which makes certain benchmarks not applicable for comparison with off-the-shelf models. However, the chosen NLU benchmarks still effectively address concerns about the potential negative impact on the models' general language understanding ability introducing by removing factual knowledge.

Table 14: Evaluation of NLU benchmarks using normalized accuracy metrics, demonstrating that separating factual knowledge during pretraining does not compromise overall model performance. A caveat is that small-scale settings constrain the scope of what we can conclude about general NLU capabilities.

| Model | Model Type | Metrics | | | | | |
| --- | --- | --- | --- | --- | --- | --- | --- |
| | | CSQA | HellaSwag | PIQA | SIQA | ARC Easy | All |
| Random Chance | - | 20.0 | 25.0 | 50.0 | 33.3 | 25.0 | 30.7 |
| OPENAI/GPT2-124M* | - | 30.3 | 29.8 | 62.5 | 40.7 | 39.5 | 40.6 |
| GPT2-124M | STANDARD | 26.5 | 26.4 | 55.3 | 39.2 | 34.2 | 36.3 |
| | LMLM | 27.9 | 26.8 | 55.1 | 39.9 | 35.0 | 37.0 |
| LLAMA2-176M | STANDARD | 26.6 | 27.0 | 55.4 | 40.4 | 33.9 | 36.7 |
| | LMLM | 26.8 | 28.2 | 55.2 | 40.2 | 35.8 | 37.2 |
| OPENAI/GPT2-355M* | - | 32.6 | 37.1 | 66.4 | 41.2 | 43.6 | 44.2 |
| GPT2-355M | STANDARD | 28.1 | 27.0 | 55.7 | 40.0 | 37.8 | 37.7 |
| | LMLM | 27.1 | 27.7 | 56.8 | 40.1 | 36.9 | 37.7 |
| LLAMA2-382M | STANDARD | 27.8 | 28.8 | 55.2 | 41.0 | 35.8 | 37.7 |
| | LMLM | 26.9 | 29.1 | 56.1 | 40.8 | 35.9 | 37.8 |

\* Models marked with an asterisk (*) are off-the-shelf models with no additional training.

## D.2 ENTITY FREQUENCY ANALYSIS IN THE KNOWLEDGE DATABASE

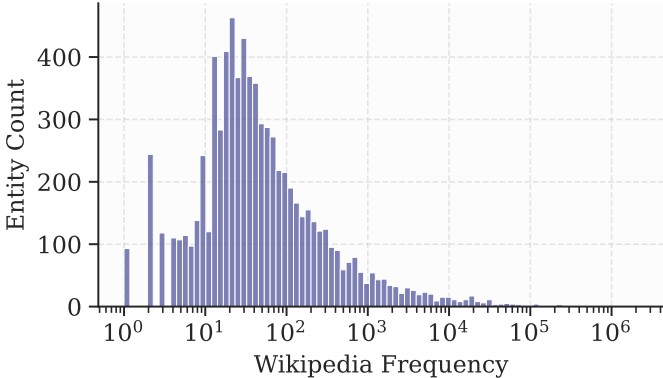

Figure 11: Frequency distribution of database entities matched to Wikipedia entries. The database spans a wide range of entity frequencies, including many long-tail cases.

---

[8]https://github.com/huggingface/lighteval

Table 15: Comparison of Fuzzy Match and Prefix-Tree Decoding. Fuzzy matching, used by default in LMLM, offers higher flexibility; prefix-tree decoding is included for ablation.

| Model | Model Type | Decoding | FActScore (%) ↑ | FActScore w/o len. penalty ↑ | Facts / Response ↑ |
|---|---|---|---|---|---|
| LLAMA2-176M | STANDARD | - | 10.1 | 11.8 | 23.7 |
| | LMLM | Prefix-tree | $23.0_{-7.6}$ | $23.9_{-8.5}$ | $34.6_{+3.2}$ |
| | LMLM | Fuzzy Match | 30.6 | 32.4 | 31.4 |
| LLAMA2-382M | STANDARD | - | 14.0 | 16.6 | 32.4 |
| | LMLM | Prefix-tree | $23.5_{-8.4}$ | $31.3_{-1.4}$ | $28.1_{-5.6}$ |
| | LMLM | Fuzzy Match | 31.9 | 32.7 | 33.7 |

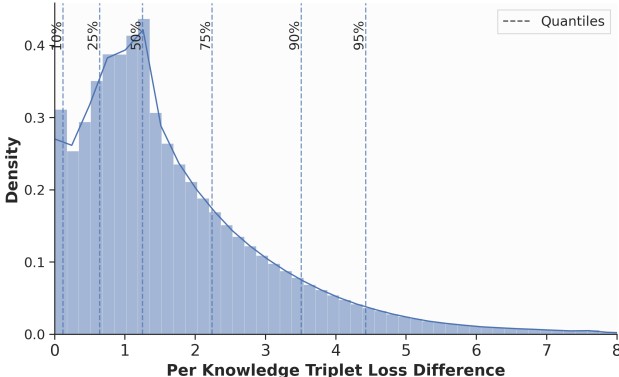

Figure 12: Distribution of per-triplet loss differences with quantile buckets used to select triplets for externalization.

We analyze the prevalence of knowledge triplets in our database by estimating how often each entity appears in Wikipedia. Using entity frequency statistics from Kandpal et al. (2023), we apply fuzzy string matching (threshold = 70) to align database entities with Wikipedia entries. Figure 11 shows the distribution of matched entity frequencies. The results indicate that our database spans both common and long-tail knowledge, with a substantial portion of entities being less frequent in the overall training corpus.

### D.3 ABLATION: UNCONSTRAINED VS. CONSTRAINED QUERY GENERATION

We compare fuzzy matching with prefix-tree constrained decoding for generating (entity, relation) queries. In the unconstrained setting, the model freely generates queries, which are then matched against the database using cosine similarity over sentence embeddings. This approach provides flexibility but may result in syntactically invalid or ambiguous queries.

In contrast, prefix-tree decoding restricts generation to valid entries encoded in a trie structure, ensuring syntactic correctness and reducing hallucinations. It is also compatible with beam search and nucleus sampling, allowing the model to explore multiple valid paths in the induced knowledge graph. See Appendix A.4 for more details.

As shown in Table 15, fuzzy matching consistently outperforms prefix-tree decoding in our setting, and is therefore used in all reported experiments. While prefix-tree decoding offers stronger structural guarantees, we find its diversity can be overly constrained by a relatively small database. As both the database and model scale, we expect the benefits of structured decoding to become more pronounced.

### D.4 ADDITIONAL RESULTS ON KNOWLEDGE-EDITING BENCHMARKS

To further demonstrate that LMLM can edit facts through database-level updates alone, we evaluated it on the multihop counterfactual subset of MQUAKE-REMASTERED (Zhong et al., 2024a), the audited version of MQUAKE (Zhong et al., 2024b). We use the full CF-6334 subset, randomly split into a train set (5334 examples) and a 1k edit eval set.

We first instruction-tune the pretrained LMLM on the train split with the *original* database for 5 epochs, so that the pretrained model learns the QA format and uses a chain of lookup calls to answer multi-hop questions. Then we evaluate edit accuracy on the 1k held-out examples using the *edited* database. For this rebuttal experiment, we follow Zhong et al. (2024a) and use the golden knowledge graph provided by the benchmark as the external database. Although this is an upper-bound setting, it is consistent with prior knowledge-editing methods that assume access to the original and edited knowledge triplets.

As shown in Table 16, LMLM outperforms all knowledge-editing baselines, confirming that factual updates can be performed through LMLM *database edits alone*, without modifying any model parameters.

Table 16: Edit accuracy on the MQuAKE-Remastered CF-6334 benchmark. Baseline results for *Vicuna-7B-v1.5* are copied from Zhong et al. (2024a) (1000-edit setting) and compared with LMLM using a LLAMA2-382M base model.

| Method | Edit Accuracy ↑ |
|---|---|
| MeLLo (Zhong et al., 2024b) | 19.27 |
| GWalk (Zhong et al., 2024a) | 61.79 |
| **LMLM (Ours)** | **69.60** |

### D.5  ANNOTATION, TRAINING, AND INFERENCE COST

To study the trade-off between LMLM's advantages and computational cost, we report data annotation, pre-training, and inference costs in Table 17. LMLM requires a one-time data annotation step to build the database and insert lookup calls, corresponding to a single inference pass of the annotator model over the pretraining corpus ($5 \times 10^{19}$ FLOPs). During pretraining, compute increases moderately ($1.72\times$ FLOPs, $1.82\times$ H100-days), mainly due to token overhead from inlined lookup calls. At inference time, generating and executing lookup calls introduces additional latency ($2.21\times$), which we have not yet optimized. This overhead is similar to that observed in current agentic LLMs that call external tools and may be reduced through system-level optimizations. Overall, the main overhead lies in pretraining compute.

In Figure 8, we extend the trade-off analysis between storing knowledge in parameters versus offloading it to the external database. We now include the corresponding compute budgets. As shown in Table 18, increasing the offloading ratio consistently improves language modeling (lower NLL) and factual precision with only a modest increase in compute.

Table 17: Compute comparison between standard pretraining and LMLM, including database construction, pretraining cost, and inference latency. Database construction is a one-time cost. Measurements use a knowledge-intensive corpus and represent a worst-case estimate.

| Method | Database Construction (FLOPs) | Training Cost (H100-days) | Training Cost (FLOPs) | Inference Latency (ms/token) |
|---|---|---|---|---|
| Standard Pretraining | N/A | 4.80 (1.0×) | $3.67 \times 10^{19}$ (1.0×) | 9.53 |
| LMLM-382M (Ours) | $5.51 \times 10^{19}$ (one-time) | 8.72 (~1.82×) | $6.32 \times 10^{19}$ (~1.72×) | 21.08 (~2.21×) |

### D.6  ANALYSIS OF LOOKUP FAILURE CASES

We first conducted additional analyses to understand how lookup failures occur and how often they happen. From 50 manually inspected T-REx examples, 64% were exact matches. The remaining failures arose from ambiguous entities/relations (6%), missing database entries (10%), multiple matches (2%), or insufficient context in the T-REx completion setting (18%). Overall, the lookup performs reliably, with the remaining errors mostly coming from limited coverage of the database.

Current fallback behavior either returns the top-1 match or outputs "unknown", and the model is not yet trained to handle these cases robustly. To improve robustness, future work will explore RL training to enable the model to reason, verify, and backtrack when the queried external knowledge is unavailable.

Table 18: Extended trade-off analysis (Figure 8): increasing the knowledge offload ratio reduces NLL with modest additional training cost.

| Knowledge Offload Ratio | Training GPU Hours (H100-days) ↓ | NLL ↓ | FactScore ↑ |
|---|---|---|---|
| 0% (Standard LLM)[†] | ∼2–3 | 8.50 | 10.1 |
| 25% | 2.48 | 7.71 | 17.2 |
| 50% | 2.91 | 7.43 | 26.1 |
| 75% | 4.10 | 7.19 | 27.1 |
| 90% | 4.31 | 6.98 | 29.2 |
| 100% (LMLM)[†] | ∼5–6 | 6.70 | 30.6 |

[†] Those training runs were launched on slightly different GPU configurations, so their training GPU hours are not directly comparable.

## D.7 QUALITATIVE RESULTS

To complement our quantitative results, we present qualitative examples comparing outputs from LMLM, STANDARD, and off-the-shelf models. As shown in Table 20, 21 and 22, LMLM produces concise, factually grounded responses by leveraging external knowledge, whereas standard and off-the-shelf models often include hallucinated content. These examples illustrate how knowledge offloading enables LMLM to maintain factual precision, further supporting our central finding (Figure 4) that LMLMs achieve more with less by scaling efficiently through externalized knowledge.

Table 19: Qualitative examples of triplets selected for externalization at different quantile buckets of per-triplet loss difference. Higher quantiles correspond to facts that are harder for STANDARD to memorize (long-tail or specific knowledge), while lower quantiles correspond to easier or contextually obvious facts. The triplet of interest in each example is highlighted in color.

| Delta Loss | Quantile | Example |
|---|---|---|
| 3.31 | 75%-100% | Joseph Charles "Joe" Avellone III, M.D. (born `<|db_start|>` Joseph Charles Avellone III`<|sep|>` Birth Date`<|db_retrieve|>` September 29, 1948`<|db_end|>` September 29, 1948) is an `<|db_start|>` Joseph Charles Avellone III`<|sep|>` Nationality`<|db_retrieve|>` American`<|db_end|>` American `<|db_start|>` Joseph Charles Avellone III`<|sep|>` Occupation`<|db_retrieve|>` medical doctor, businessman, politician`<|db_end|>` medical doctor, businessman, and politician from `<|db_start|>` Joseph Charles Avellone III`<|sep|>` State of Residence`<|db_retrieve|>` Massachusetts`<|db_end|>` Massachusetts...He then worked as CEO of biomedical company `<|db_start|>` Joseph Charles Avellone III`<|sep|>` CEO Of`<|db_retrieve|>` Veritas Medicine`<|db_end|>` Veritas Medicine ... |
| 3.23 | 75%-100% | Elsa van Dien...Her thesis, supervised by `<|db_start|>` Elsa van Dien`<|sep|>` Thesis Supervisor`<|db_retrieve|>` Donald Menzel`<|db_end|>` Donald Menzel , discussed the Stark effect in the Balmer lines of early type stars. |
| 1.41 | 50%-75% | Kermia albicaudata is a `<|db_start|>` Kermia albicaudata`<|sep|>` Species Type`<|db_retrieve|>` sea snail`<|db_end|>` species of sea snail, a marine gastropod mollusk in the `<|db_start|>` Kermia albicaudata`<|sep|>` Family`<|db_retrieve|>` Raphitomidae`<|db_end|>` family Raphitomidae.... Distribution. This species occurs in the `<|db_start|>` Kermia albicaudata`<|sep|>` Geographical Distribution`<|db_retrieve|>` Persian Gulf`<|db_end|>` Persian Gulf . |
| 0.82 | 25%-50% | Havering Country Park... The land was purchased by the `<|db_start|>` Havering Country Park`<|sep|>` Purchased By`<|db_retrieve|>` Greater London Council`<|db_end|>` Greater London Council and opened to the public in `<|db_start|>` Havering Country Park`<|sep|>` Opening Year`<|db_retrieve|>` 1975`<|db_end|>` 1975, with ownership transferring to `<|db_start|>` Havering Country Park`<|sep|>` Ownership Transferred To`<|db_retrieve|>` Havering Council`<|db_end|>` Havering Council in `<|db_start|>` Havering Council`<|sep|>` Ownership Year`<|db_retrieve|>` 1986`<|db_end|>` 1986. |
| 0.003 | 0%-25% | Kagoshima District, Kagoshima is a district located in `<|db_start|>` Kagoshima District, Kagoshima`<|sep|>` Located In Prefecture`<|db_retrieve|>` Kagoshima Prefecture, Japan`<|db_end|>` Kagoshima Prefecture, Japan . |

Table 20: Qualitative examples of generated outputs from models with STANDARD and LMLM settings.

| Model | Generated Output |
|---|---|
| LLAMA2-176M-LMLM | Ko Itakura is the `<|db_start|> Ko Itakura<|sep|> Position<|db_retrieve|>` center-back, defensive `midfielder<|db_end|>` **center-back and defensive midfielder** for `<|db_start|> Ko Itakura<|sep|> Current Team<|db_retrieve|>` Japan `national team<|db_end|>` **Japan's senior squad in the** `<|db_start|> Ko Itakura<|sep|> League Participation<|db_retrieve|>` 2022 `FIFA World Cup<|db_end|>` **2022 FIFA World Cup.** He was born on `<|db_start|> Ko Itakura<|sep|> Birth Date<|db_retrieve|>` `January 27, 1997<|db_end|>` **January 27, 1997.**

Career. On `<|db_start|> Ko Itakura<|sep|> Joined Club<|db_retrieve|>` `Kawasaki Frontale<|db_end|>` **July 30, 2015,** it was announced that Itakura joined JFL club Kawasaki Frontale from their youth system. In `<|db_start|> Ko Itakura<|sep|> Moved To<|db_retrieve|>` `Groningen<|db_end|>` **August 2016 he moved to Groningen.** On `<|db_start|> Ko Itakura<|sep|> Debut Date<|db_retrieve|>` `June 17, 2019<|db_end|>` **June 17, 2019, Itakura debuted against** `<|db_start|> Ko Itakura<|sep|> Debut Opponent<|db_retrieve|>` `Uruguay<|db_end|>` **Uruguay in an international friendly match.** After playing only one game for Groningen during his debut season, Itakura left the club at the end of the year. |
| LLAMA2-176M-STANDARD | Ko Itakura is the name given to an ancient Japanese settlement in what is now part of Shimane Prefecture, Japan. The site was discovered by archaeologists from Kyoto University and has been excavated since 1978.

The first written reference to this place appears in "Nihon Kōki", which dates back to the year 604 AD (Kumano no Kan'i), but it seems that there were already settlements at this location before then. In the early Heian period, the area around modern-day Shimonoseki City became known as , meaning "the land between rivers". During the Kamakura period, the region came under control of the Ashikaga clan who built their capital on Mount Hiei near present-day Naka Ward. After the fall of the Ashikaga shogunate, they lost control over much of eastern Japan, including the area surrounding modern-day Shimane prefecture. However, after the Battle of Sekigahara, Tokugawa Ieyasu established himself as lord of most of northern Japan with his domain centered on Matsue Castle. This castle later fell into disrepair during the Edo Period, and |

Table 21: Qualitative examples of generated outputs from models with STANDARD and LMLM settings.

| Model | Generated Output |
|---|---|
| LLAMA2-382M-LMLM | Ko Itakura is the `<\|db_start\|> Ko Itakura<\|sep\|> Position<\|db_retrieve\|> center-back, defensive midfielder<\|db_end\|>` center-back and defensive midfielder for `<\|db_start\|> Ko Itakura<\|sep\|> Team<\|db_retrieve\|> Japan national team<\|db_end\|>` Japan's national team. He was born in `<\|db_start\|> Ko Itakura<\|sep\|> Birthplace<\|db_retrieve\|> Yokohama<\|db_end\|>` Yokohama on `<\|db_start\|> Ko Itakura<\|sep\|> Birth Date<\|db_retrieve\|> January 27, 1997<\|db_end\|>` 27th January 1997. His father is from `<\|db_start\|> Ko Itakura<\|sep\|> Father's Origin<\|db_retrieve\|> Japanese<\|db_end\|>` Japanese and his mother is from `<\|db_start\|> Ko Itakura<\|sep\|> Mother's Origin<\|db_retrieve\|> Japanese<\|db_end\|>` Japanese. When he was young, he played baseball but after watching an exhibition match against a professional baseball club, he decided to become a footballer. In `<\|db_start\|> Ko Itakura<\|sep\|> Joined Club Year<\|db_retrieve\|> Kawasaki Frontale<\|db_end\|>` 2013, he joined J1 League side Kawasaki Frontale. However, he could not play many matches behind Shusaku Nishikawa until `<\|db_start\|> Ko Itakura<\|sep\|> First Match as Starter<\|db_retrieve\|> Uruguay<\|db_end\|>` September when he debuted at right back against Uruguay. After that, he became a regular player under manager `<\|db_start\|> Ko Itakura<\|sep\|> Manager Under Whom Became Regular Player<\|db_retrieve\|> Japan national team<\|db_end\|>` Shinji Ono. On `<\|db_start\|> Ko Itakura<\|sep\|> Debut Date<\|db_retrieve\|> June 17, 2019<\|db_end\|>` 17 June 2019, he debuted with Japan national team against Chile during the `<\|db_start\|> Ko Itakura<\|sep\|> Competition Debut<\|db_retrieve\|> Uruguay<\|db_end\|>` 2019 Copa América. Career statistics. "Updated to end of 2018 season". National team career. In August 2016, Itakura was elected to the `<\|db_start\|> Ko Itakura<\|sep\|> U-23 Selection<\|db_retrieve\|> Japan U-20 national team<\|db_end\|>` Japan U-20 national team for the `<\|db_start\|> Japan U-20 national team<\|sep\|> Tournament Participation<\|db_retrieve\|> 2017 U-20 World Cup<\|db_end\|>` 2017 U-20 World Cup. At this tournament, he played all 4 matches as left back of three back defense. In May 2019, he was selected for the `<\|db_start\|> Ko Itakura<\|sep\|> Senior Squad Selection<\|db_retrieve\|>` |
| LLAMA2-382M-STANDARD | Ko Itakura is the first album by Japanese singer-songwriter Koda Kumi, released on November 19, 2003 under Sony Music Japan label. The album was reissued in March 2004 with two bonus tracks: "Kimi no Uta" and "Tell Me A Bio". |

Table 22: Qualitative examples of generated outputs from different off-the-shelf models.

| Model | Generated Output |
|---|---|
| OPENAI/GPT2-124M | Ko Itakura is the author and editor of *The Art Of Anime*, which has been published by Kadokawa since 2013 (and was also translated into English in 2015). I'm not sure if I can say much more about this book than what you've said before but it's pretty good! This manga really does have an interesting story arc that makes for some great reading experience as well as being very entertaining to read on its own terms!! |
| OPENAI/GPT2-355M | Ko Itakura is an American author, illustrator and animator who has worked on many anime series including: Naruto, One Piece (one of the most popular manga in Japan), Bleach, Dragon Ball Z, Ghost in The Shell, Gintama.

He also created characters for various games such as Final Fantasy VII, Pokémon X/Y, or Pokemon Omega Ruby/Alpha Sapphire.

The story takes place during World War II when Japanese soldiers are sent to fight against Nazi Germany's forces under General Yamamoto at Kure Castle near Tokyo.

After defeating them all except one—Genosha-Kaiji from Earth Kingdom fame—they return home with their families but find that there was no sign left behind by those killed fighting alongside them; instead it seems like everything had been destroyed!

Now this isn't exactly what you'd expect if your favorite character were actually dead... but then again maybe he wasn't?

Well I'm sure we'll get into more details about him later so stay tuned... |
| PYTHIA-1B | Ko Itakura is the name of my father.¨

Ï don't know him,¨said the man, ¨but I can tell you that he was an excellent swordsman and a good fighter. He had many friends in the army who were killed by bandits or robbers when they tried to capture them. They would have been glad if someone like him could help them escape from their captors. But it's not possible for anyone with his skill to be captured alive. So we must find out where he lives so that we may rescue him.¨

The man went on: ¨He has lived here all these years because there are no other people around. The only thing that keeps him going is the hope that someday some day he will return home. And now he wants to see you again. Please come this way.¨

They walked along the road until they came to a small house which looked as though it might belong to one of the villagers. A woman opened the door and invited them inside. She led them into a room full of bookshelves. There were several large tables covered with maps and charts. On each table sat a young girl about ten years old. Her hair was long and black and her eyes were bright blue. She wore a white dress and carried |
| LLAMA2-7B | Ko Itakura is the author of *The Artist's Way* (4.06 avg rating, 12 ratings, 3 reviews), *The Artist's Way Workbook: A Course in Discovering and Recovering Your Creative Self* (5/5(7)).

*The Artist's Way* by Julia Cameron – Book Review — The Guardian

Julia Cameron has been an active artist for more than thirty years. She was born in London but now lives in New York City with her husband, Martin. They have three children together. In addition to writing books on creativity, she also writes plays and musicals. Her latest book, *Finding Water*, will be published this fall.

She is currently working on another novel called *The Last Days Of Summer*. This article contains affiliate links which means that if you click through from here we may receive compensation at no extra cost to yourself! We only recommend products or services that we believe are useful for our readers so please feel free to check them out before making any purchases :) Thank you for reading our blog post today!! If there's one thing I know about myself it's how much I love reading blog posts like these ones because they always make my day better when someone |

Table 23: Example of input and output from the RAG model. The model is prompted using retrieved documents and a user query.

| Field | Content |
|---|---|
| **Prompt Template** | `Answer the question or complete the prompt based on the given document.  The following are given documents:` `[RETRIEVED_DOCUMENTS] \n [USER_QUERY]` |
| **Retrieved Documents** | `"Kang Ji-hwan" is a South Korean actor born Jo Tae-gyu` `on March 20, 1977.  He began in musical theatre before` `transitioning to television and film, with notable roles` `in dramas like "Be Strong, Geum-soon!", "Capital Scandal",` `and "Hong Gil-dong".` `"90 Days, Time to Love" is a 2006 South Korean TV series` `starring Kang Ji-hwan, centered on a professor and` `scriptwriter who rekindle a romance after discovering they` `are cousins.` `Kang also produced and starred in the musical "Cafe In",` `becoming the first Korean actor to perform at Tokyo Globe` `Theatre.` |
| **User Query** | *Tell me a bio of Kang Ji-hwan.* |
| **Model Input** | `Answer the question or complete the prompt based on the` `given document.  The following are given documents:` `"Kang Ji-hwan" is a South Korean actor born Jo Tae-gyu` `on March 20, 1977.  He began in musical theatre before` `transitioning to television and film, with notable roles` `in dramas like "Be Strong, Geum-soon!", "Capital Scandal",` `and "Hong Gil-dong".` `"90 Days, Time to Love" is a 2006 South Korean TV series` `starring Kang Ji-hwan, centered on a professor and` `scriptwriter who rekindle a romance after discovering they` `are cousins.` `Kang also produced and starred in the musical "Cafe In",` `becoming the first Korean actor to perform at Tokyo Globe` `Theatre.` `Tell me a bio of Kang Ji-hwan.` |
| **Model Output (OPENAI/GPT2-355M)** | Kang Ji-hwan is known primarily for portraying characters with complex emotions like "the heartbreak," which can be seen throughout most drama shows nowadays. His character arc often involves both romantic relationships and difficult situations where people have conflicting feelings about their own lives. For example during this season we see how complicated it is for two men trying desperately not get divorced because there isn't enough time left before graduation! What do you think? Do your favorite actors play these kinds/characters differently than others? Let us know what kind(ies?) I missed out below... |
| **Model Output (LLAMA-3.2-1B-INSTRUCT)** | Kang Ji-hwan is a renowned South Korean actor known for his versatility and range in various genres.
**Early Life**
Kang Ji-hwan was born on March 20, 1977, in Busan, South Korea. His full name is Jo Tae-gyu.
**Career Beginnings**
He started his career in musical theater before transitioning into film and television. Kang debuted in musical performances in 2001 with stints in productions like "The Rocky Horror Show" and "Grease".
**Breakthrough Roles**
His breakthrough came in 2005 with the drama "Be Strong, Geum-soon!". He later starred in "Capital Scandal", "Hong Gil-dong", and "90 Days, Time to Love", showcasing his ability to play complex characters.
**Film Career**
Kang ventured into cinema with movies like "Rough Cut" (2008) and "My Girlfriend Is an Agent" (2009).
**Recognition**
Throughout his career, Kang has received numerous awards and nominations, solidifying his status in Korean entertainment. |

## E    LIMITATIONS AND FUTURE WORK

**Toward Efficient Scaling via More Complex Knowledge Structures**    As discussed in Section 6, extending our method beyond entity-level knowledge remains an open challenge. Future work includes exploring more flexible syntax and richer structured representations, which could improve robustness and applicability in realistic post-training environments.

**Evaluation beyond Factuality**    Evaluating instruction-following or reasoning ability is important for understanding a pretraining paradigm. At the small scale, however, it is challenging to meaningfully evaluate later-stage capabilities (e.g. instruction following or RL-based reasoning), which is a limitation we acknowledge. Among current open-source small models (e.g., Pythia (Biderman et al., 2023), Olmo (Groeneveld et al., 2024), TinyLlama (Zhang et al., 2024a), SmolLM3 (Bakouch et al., 2025)), only SmolLM3-3B performs mid-training and post-training evaluation at scale. It remains standard practice for base models to be evaluated through pretraining NLU tasks and perplexity, which serve as reliable indicators for data quality and pretraining effectiveness. Within NLU, HellaSwag and CommonsenseQA emphasize commonsense reasoning, while PIQA includes knowledge and linguistic understanding. We agree that it remains an exciting and open question how to design base models that are inherently RL-able or naturally capable of tool use. Within the scope of this paper, we focus on knowledge memorization during pretraining. We leave it for future work.

**Compatibility with Instruction Tuning**    It is important to assess whether the observed advantages of LMLM persist after instruction tuning. Our mechanism operates entirely through data-level supervision. In principle, the same LMLM-style entity annotation and masked next-token prediction loss can be applied to knowledge-intensive instruction data, while leaving non-knowledge instructions unchanged. This makes the approach compatible with standard instruction-tuning pipelines.

More importantly, instruction tuning primarily aligns the model's response behavior, teaching it to follow user instructions and handle diverse task formats, rather than to acquire new factual knowledge (Sanh et al., 2022; Zhou et al., 2023). LMLM, in contrast, controls whether factual knowledge is stored parametrically or offloaded to the external database. These objectives are complementary. Therefore, we expect LMLM's advantages to be preserved under instruction tuning. Further experiment leaves for future work.

**Annotation Cost, Token Overhead, and Safety Risks**    LMLM requires a one-time data annotation step to build the database and insert lookup calls, corresponding to a single inference pass of the annotator model over the pretraining corpus ($5 \times 10^{19}$ FLOPs). During pretraining, compute increases moderately ($1.72\times$ FLOPs, $1.82\times$ H100-days), mainly due to token overhead from inlined lookup calls. At inference time, generating and executing lookup calls introduces additional latency ($2.21\times$), which we have not yet optimized. This overhead is similar to that observed in current agentic LLMs that call external tools, and might be reduced through system-level optimizations.

We are also aware of the potential risks of introducing annotations directly into pretraining data. Additional filtering and safeguards will be important for mitigating safety-related or malicious-content risks.

**Challenges in Applying LMLM to New Domains**    We identify two main challenges when extending LMLM to new domains such as books or scientific domains. First, current knowledge triplet format is limited and can not capture all types of factual knowledge. Second, in scientific domains where entity-level knowledge is the main focus, standard language models inevitably hallucinate while LMLMs offer the benefits of controllability and interpretability. To enhance robustness, future work will explore RL training to enable the model to reason, verify, and backtrack when the queried external knowledge is unavailable.

**Small-Scale Constraints**    We acknowledge that, given the limited computation budget, it is difficult to fully assess how well smaller LMLM models preserve general NLU abilities.

**Remain Parametric Knowledge**    Models still require some factual knowledge to conduct effective queries.

## F   LLM USAGE

We used LLMs as general-purpose assistive tools to improve clarity, grammar, and readability of the paper. LLMs were also used for creating tables and figures, but not for research ideation, experiment design, or substantive writing. All research contributions, methodology, experiments, and analyses were conceived, implemented, and written by the authors.

