# OpenReview forum: "Pre-training Limited Memory Language Models with Internal and External Knowledge"
_ICLR.cc/2026/Conference — ICLR 2026 Poster_

### Official Review · Reviewer_pSTT · 2025-10-27

**Soundness:** 3
**Presentation:** 3
**Contribution:** 3
**Rating:** 8
**Confidence:** 3

**Summary:**

This paper introduces a method for training language models that query an external database for generating factual information, instead of relying on internal parametric knowledge. The method involves annotating the pre-training data to identify factoids; constructing a knowledge base; and then training LMs to query the database to complete factoids. The method is evaluated in terms of perplexity; factuality (e.g. FActScore); and on applications like machine forgetting.

**Strengths:**

- In my opinion, this paper proposes a good idea for addressing an important problem. Externalizing factual information could make LMs more interpretable and controllable, which could make LMs more useful in domains where these attributes are important.

- The idea is conceptually simple, but not simple to execute. This paper does an impressive job implementing all of the different components (pre-processing, generating the database, search augmentation, etc.).

- The empirical evaluations are fairly thorough. The experiments show that perplexity is improved (relative to larger models), knowledge can be deleted, language understanding performance is still good. The paper also shows that there are improvements to downstream factuality metrics, like FactScore, and reports some interesting analysis into how much information is still retained, and the optimal "offloading ratio".

- The results are generally strong, with LMLM achieving better perplexity, higher FactScore, decent NLU, and high forgetting quality without loss of utility.

- The writing and presentation are generally very clear.

**Weaknesses:**

- The paper suggests that this method can make it easier to update facts, but there do not seem to be any experiments on knowledge editing benchmarks (e.g. MQuake [1]), except for the unlearning experiment.

- Some design choices are not thoroughly validated. For example, in section 3.1, it is not clear whether it is necessary to do the intermediate filtering step (which involves fine-tuning a corrector model) in the entity annotation phase. There is also no evaluation of the annotator performance. However, given the many different components of this pipeline, I think it is reasonable to omit some of these ablations, given the good downstream results.

- The models are trained only on Wikipedia. There might be considerable challenges to extending this approach to other domains, e.g. books, or scientific domains. The authors acknowledge this limitation, but it would still be interesting to see some experiments on other domains.

- The experiments are limited to very small models. This could be a reasonable limitation, given limited computation budget, but it makes it more difficult to interpret the results. For example, the NLU metrics (table 12) are all very low, so it is hard to really appreciate how well these models preserve NLU abilities.

- Minor comment: The paper shows evidence that this method reduces factual memorization, but some factual knowledge will always be needed to issue good queries. For example, in the example in the appendix, the model needs to know that Ko Itakura is an athlete to look up "Position" and "Team". This is a minor point, but in my opinion it is worth mentioning, as it makes it could qualify the claims about knowledge editing and unlearning.

- Overall, I think most of these weaknesses are acceptible limitations, given the scope of this work, but I would have appreciated a more extensive discussion of the limitations--especially more concrete discussion of the limitations of applying this approach to non-Wikipedia domains.



[1] MQuAKE: Assessing Knowledge Editing in Language Models via Multi-Hop Questions. Zhong et al., 2023.

**Questions:**

- In section 4.1, does the database include triplets from the held-out set? How many entities in the held-out set are not in the database?

- Can you clarify the setup of the LMLM model for the TOFU experiments (sec. 4.3)? It seems that the LMLM model in this section is a version of Llama-3.2-1B-Instruct that has been fine-tuned on additional data to learn the lookup mechanism. Is there any reason to use this approach, rather than evaluating one of the models that was fine-tuned from scratch in the previous section? Alternatively, could you report NLU results for this fine-tuned model?

Minor comments:

- In Figure 5, I think it would be clearer to label the x-axis "Model configuration" or "Model architecture".

---

> ### Author Response · Authors · 2025-11-21
>
> Thanks for the thoughtful and encouraging feedback. We appreciate your recognition of the novelty of our idea, the implementation effort, extensive evaluation and the generally strong results. We also value your clear understanding of both the scope and limitations of this work. Below, we address each concern.
>
> ---
> ### **W1: Additional results on knowledge-editing benchmarks (e.g., MQuake)**
>
> To further demonstrate that LMLM can edit facts through database-level updates alone, we evaluated it on the multihop counterfactual subset of MQuAKE-Remastered [1], the audited version of MQuAKE [2]. We use the full CF-6334 subset, randomly split into a train set (5334 examples) and a 1k edit evalset.
>
> We first instruction-tune the pretrained LMLM (LLaMA2-382M) on the train set with the *original* database for 5 epochs, so that the pretrained model learns the QA format and uses chain of lookup calls to answer multi-hop questions. Then we evaluate edit accuracy on the 1k held-out examples using the *edited* database. For this rebuttal experiment, we follow [1] and use the golden knowledge graph provided by the benchmark as the external database. Although this is an upperbound setting, it’s consistent with prior knowledge editing methods that assume access to the original and edited knowledge triplets.
>
> | Method | Edit Accuracy↑|
> |--------|-----------|
> | MeLLo [2] | 19.27 |
> | ICE | OOM |
> | IKE | OOM |
> | PokeMQA | – |
> | DeepEdit | <1 |
> | GWalk [1] | 61.79 |
> | **LMLM (Ours)** | **69.60** |
>
> > *Note 1*: Baseline results are from MQuAKE-Remastered-CF-6334 1000-edit setting in Table 4 of [1], which use Vicuna-7B-v1.5 as the base model. Our experiment uses LMLM-LLaMA2-382M, a much smaller model, but achieves the highest performance.
>
> > *Note 2*: Most of LMLM’s remaining errors come from contamination between overlapping edits in the benchmark.
>
>
> LMLM outperformed the knowledge editing baselines, confirming that factual updates can be performed through *LMLM database edits alone* without modifying model parameters. We have added these results to the Appendix D.4. Thanks for the suggestion.
>
>
> [1] Zhong, Shaochen, et al. "MQuAKE-Remastered: Multi-Hop Knowledge Editing Can Only Be Advanced With Reliable Evaluations." ICLR 2025.
>
> [2] Zhong, Zexuan, et al. "Mquake: Assessing knowledge editing in language models via multi-hop questions." EMNLP 2023.
>
> ---
>
> ### **W2: More details on the annotation pipeline ablation**
>
> We appreciate the reviewer’s suggestion. This ablation was omitted in the submitted version, and we now provide the details below.
>
> We evaluated different annotator training configurations on a GPT-4 annotated seed evalset containing 100 SQuAD passages and 100 Wikipedia articles, which were cleaned to ensure reliable supervision signals. The *Annotator (ours)* was lora finetuned for 10 epochs on a mixed dataset of 1 k SQuAD and 1 k Wikipedia examples with the intermediate filtering step described in Sec 3.1. Two ablation baselines were compared: *- w/o data mixing* that trained only on 19k SQuAD while keeping total compute constant, and *- w/o filtering* that trained on the same mixed data but without the filtering stage.
>
>
> | Model                                                                   | wiki_org   | wiki_db-avg | squad_org   | squad_db-avg |
> | ---------------------------------------------------------------------- | ----------- | ------------ | ----------- | ------------ |
> | **Annotator (ours)** | **3.33e-3** | **7.85e-2**  | **2.93e-3** | **1.22e-1**  |
> | *- w/o data mixing*              | 4.21e-3     | 1.18e-1      | 3.67e-3     | 1.81e-1      |
> | *- w/o filtering*  | 3.38e-3     | 8.21e-2      | 3.31e-3     | 1.30e-1      |
> > *Note*: Loss is decomposed into the original corpus and the average loss on lookup calls.
>
> As shown, the mixed-data and filtering design achieves the lowest losses across both domains, confirming that both steps contribute meaningfully to annotation quality. We have added this ablation detail to the updated Appendix A.1.
>
> ---
>
> ### **W3: Expanded discussion of limitations**
>
> We have substantially extended the limitations discussion in the Appendix E, covering:
>
> **Extending to other domains:**
> We identify two main challenges when extending LMLM to new domains such as books or scientific domains.
> First, current knowledge triplet format is limited and can not capture all types of factual knowledge, as detailed discussed in sec.6.
> Second, in scientific domains where entity-level knowledge is the main focus, such as in materials science [1], standard language models inevitably hallucinate while LMLMs offer the benefits of controllability and interpretability. To enhance robustness, future work will explore RL training to enable the model to reason, verify, and backtrack when the queried external knowledge is unavailable.

---

> > ### Author Response · Authors · 2025-11-21
> >
> > **Constraints due to current small model scale:**
> > We acknowledge that, given the limited computation budget, it is difficult to fully assess how well smaller LMLM models preserve general NLU abilities. Previous work [2] shows that pretraining on small size (e.g., 150M parameters) is a strong proxy for predicting performance trends at larger target scale (1B) (∼ 80% accuracy across comparisons), suggesting that our results provide reliable signals for larger-scale behavior. We have added this clarification to the caption of Table 12.
> >
> > **minor comment:**
> >  We also mention that models still require some factual knowledge to conduct effective queries, included in the updated Limitations section.
> >
> >
> > [1] Itani, Suman, Yibo Zhang, and Jiadong Zang. "The northeast materials database for magnetic materials." Nature Communications 16.1 (2025): 9415.
> >
> > [2] Magnusson, Ian, et al. "Datadecide: How to predict best pretraining data with small experiments." ICML, 2025.
> >
> > ---
> >
> > ### **Q1: Clarification of database setup**
> >
> > As mentioned in Sec. 4.1, we construct the database from the full annotated Wikipedia (including both the pretraining corpus and the 1k held-out pages used only for perplexity evaluation), which is consistent with standard practice in retrieval-augmented systems where the entire corpus serves as the external datastore. The LMLM database is used purely as a non-parametric external store.
> >
> > For perplexity evaluation on the held-out set (sec. 4.2), this does not introduce data leakage or bias because the model is never trained on the held-out annotated text.
> >
> > For factuality evaluation (sec. 4.4), there is no held-out split by design. These benchmarks (e.g., FActScore, PopQA, T-REx) are themselves constructed from Wikipedia and are intended to assess factual precision in an in-domain setting. This setup is fair for both the Standard model and the LMLM model, since both are pretrained on the same Wikipedia corpus. The difference lies only in whether factual knowledge is stored in the model parameters or offloaded externally.
> >
> > For unlearning evaluation (sec. 4.3), we additionally construct a database from the full TOFU dataset (4k QA pairs), as TOFU is synthetic and not covered by Wikipedia. We follow the original TOFU paper, which trains on the full dataset and evaluates on a subset drawn from the training data, reflecting an in-domain setting intended to measure unlearning effectiveness on knowledge that the model has previously been trained on.
> >
> >
> > ---
> > ### **Q2: Clarification of TOFU setup and additional results**
> >
> > Yes. As noted in Table 8, we include an additional warm-up finetune step for the Llama-3.2-1B-Instruct backbone so that it learns the LMLM lookup format before applying TOFU evaluation. We use this backbone for two reasons:
> >
> >  (1) TOFU is designed for instruction models, and the official benchmark reports all baselines using instruction-tuned backbones.
> >
> >  (2) TOFU is synthetic, so using an instruction model that has never seen the underlying knowledge provides a fair initialization to show that LMLM prevents knowledge internalization by design.
> >
> > We agree with the reviewer, however, that evaluating our pretrained LMLM models on TOFU is also valuable. We now include results using LMLM-LLaMA2-382M, which was trained from scratch under the LMLM paradigm. As shown below, both LMLM variants achieve effective unlearning (p-value > 0.05) without degrading model utility, consistent with the results in Section 4.3. We include this in the updated Appendix C.4.
> >
> > | Method     | model utility (-) | forget quality (p-value ↑) |
> > |------------|--------------------|-----------------------------|
> > | Finetuned | 0.60               | 1.33e-13                    |
> > | Retain (Ideal)    | 0.60               | 1.00                        |
> > | GradAscent | 0.00               | 1.94e-119                   |
> > | GradDiff   | 0.53               | 1.94e-119                   |
> > | IdkDPO     | 0.07               | 1.12e-05                    |
> > | NPO        | 0.45               | 0.14                        |
> > | SimNPO     | 0.58               | 5.01e-100                   |
> > | RMU        | 0.58               | 4.87e-10                    |
> > | LMLM (LLaMA-3.2-1B-Instruct)   | 0.55               | **0.31**                        |
> > | LMLM (LLaMA2-382M)   | 0.43               | **0.70**                        |
> > > *Note*: baseline numbers use Llama-3.2-1B-Instruct from the official TOFU repository.
> >
> > ---
> > ### **Q3: Minor: fix the x-axis label in Figure 5**
> >
> > Thanks for pointing it out. We have corrected the x-axis label in the updated version.

---

> > > ### Comment · Reviewer_pSTT · 2025-11-26
> > >
> > > Thank you for the detailed reply. I appreciate the clarifications, and I think the paper will be stronger with the expanded limitation section and the additional results (annotator analysis, MQuake knowledge editing updates, additional TOFU results). I have also read the other reviews. I still think this is a strong paper that is worthy of being accepted and I will maintain my score.

---

> > > > ### Author Response · Authors · 2025-11-26
> > > >
> > > > Thank you for the thoughtful follow-up and for maintaining your score. We’ve incorporated the revised limitations and additional results into the paper.
> > > >
> > > > Thanks again for your time. Your support and constructive comments have been very helpful in strengthening this paper further.

---

### Official Review · Reviewer_qoG2 · 2025-10-28

**Soundness:** 2
**Presentation:** 3
**Contribution:** 2
**Rating:** 4
**Confidence:** 4

**Summary:**

This paper proposes a parametric & non-parametric hybrid of language modeling based on the construction and utilization of a factual knowledge database from the pre-training stage. The proposed method, LMLM, differs from standard retrieval methods like RAG in that it aims to dissect factual knowledge from other general language competencies. Experiments on Wikipedia pretraining data show that medium-sized LMLMs, compared to standard language models, show advantages on tasks that require factual knowledge memorization and knowledge unlearning.

**Strengths:**

S1: A lot of experimental analyses are well executed and presented, supporting the core advantages of the proposed method well

S2: The limitations of the proposed method and future research directions are well stated in the discussion section

**Weaknesses:**

W1: **Limited scope of usage** - While the proposed method can be utilized in knowledge-intensive tasks, whether it can be extended to broader usage is unclear. This is because further tuning of LMLMs will likely require a similar formatting of fine-tuning data by design. For example, I'm not clear about whether the proposed method can be deployed and maintained under instruction tuning.

W2: **Brittleness of DB-style modeling of factual knowledge** - While the proposed method inherits many nice properties of symbolic modeling of factual knowledge (such as precision and interpretability), it also inherits their drawbacks. For example, as the authors mentioned, the modeling of factual knowledge as a triplet of (entity, relation, entity) does not encompass all kinds of factual knowledge. For example, the proposed method cannot cover factual knowledge like "Michael Jordan is tall" or "John gave a watch to Mary as a present". Moreover, the behavior of the model on the specific set of factual knowledge that has failed to be incorporated into the database can be more brittle compared to standard LMs, which is not thoroughly investigated in the experiments. Specifically, I believe pre-training on the Wikipedia dataset that is dominated by 'well-shaped factual knowledge' might provide an unfair advantage to LMLMs, which can behave much more brittle in a realistic corpus.

W3: **Lacking comparison of train/inference computational cost** - As the proposed method necessitates the enumeration over the whole pretraining data for database construction, the overall training compute required is expected to be much larger than standard LM pretraining. Moreover, as it requires access to the database, inference cost will be increased, too. However, no quantitative information about the additionally required computing is provided in the main text. It would be great to see this information (for example, in wall-clock time or FLOPs) to investigate the trade-off between its advantage and computational burden.

**Questions:**

Q1: I couldn't find the comparison of the unlearning performance of LMLM and RAG-based models. I believe this comparison is crucial because it is fairer to compare the unlearning performance of two methods that have direct access to a pre-stored database.

Q2: How did you handle the 'failure cases', for example, when the model generated a syntactically infeasible database query or a query that the database cannot answer, and how often did it happen? Or, did you preclude such possibilities by employing a constrained decoding methods?

---

> ### Author Response · Authors · 2025-11-21
>
> Thanks for your insightful comments and suggestions. We appreciate the feedback, which helps further strengthen the paper. In the revised version, we expanded the discussion of limitations and added the requested experimental results in the Appendix. Specifically, we address each concern below.
>
> ---
> ### **W1: whether the proposed method can be deployed and maintained under instruction tuning**
>
> We agree that evaluating whether the observed advantages persist under instruction tuning is important for understanding the generality of this paradigm. Our mechanism operates purely through data-level supervision. In principle, it can be made compatible with instruction tuning by consistently applying the same LMLM-style entity-level annotation and masked next token prediction loss to knowledge-intensive instruction data, while leaving other instruction data unchanged.
>
> More importantly, instruction tuning mainly teaches models to respond to diverse end tasks and user interactions, exposing the knowledge and capabilities learned during pretraining [1-2]. It aligns response behavior rather than acquiring new knowledge, while LMLM controls whether factual knowledge is offloaded and stored externally. These objectives are complementary. Therefore, we expect it to remain compatible under instruction tuning. We have added this discussion to the updated Limitations section in Appendix E.
>
>
>
> [1] Zhou, Chunting, et al. "Lima: Less is more for alignment." Advances in Neural Information Processing Systems 36 (2023): 55006-55021.
>
> [2] Sanh, Victor, et al. "Multitask prompted training enables zero-shot task generalization." arXiv preprint arXiv:2110.08207 (2021).
>
> ---
>
> ### **W2: Brittleness of DB-style modeling of factual knowledge**
>
> > the behavior of the model on the specific set of factual knowledge that has failed to be incorporated into the database can be more brittle compared to standard LMs
>
> Our work focuses on offloading atomic, entity-level factual knowledge using a simple and effective separation pipeline. We agree that the current triplet does not capture all types of factual knowledge, as mentioned in the limitation section. We view this as a reasonable starting point for offloading the subset of factual knowledge where database storage provides clear advantages of precision and controllability, while all other knowledge remains parametric.
>
> As discussed in Q2, lookup failures do occur due to missing entries (≈10% on the T-REx benchmark). In these cases, the model returns the top-1 match, outputs “unknown,” or falls back to standard parametric prediction. We acknowledge that LMLM is not yet explicitly trained to handle missing facts robustly.
>
> Overall, explicit knowledge separation offers advantages in settings where controllability and verifiability are important. Even with this partial coverage, our empirical results show that LMLMs improve pre-training efficiency (sec 4.2), enable instant unlearning (sec 4.3), and improve factuality (sec 4.4).
>
> Improving database coverage and training the model to handle query failure is a natural and important direction for future work (e.g. RL training to teach the model to reason, verify, and backtrack when lookup query fails). We have expanded this discussion in the revised Appendix E.

---

> ### Author Response · Authors · 2025-11-21
>
> ### **W3: Train/inference computational cost**
>
> Thank you for pointing this out. To study the trade-off between LMLM’s advantages and computational cost, we report data annotation, pre-training and inference costs below.  LMLM requires a one-time data annotation to build the database and insert the lookup calls, which corresponds to a single inference pass of the annotator model over the pretraining corpus (5 × 10¹⁹ FLOPs). During pretraining, the compute increases moderately (1.72× FLOPs, 1.82× H100-days) mainly due to token overhead from inlined lookup calls. At inference, generating and executing lookup calls introduces a latency increase (2.21×) which we have not yet optimized. This overhead is similar to that observed in current agentic LLMs that call external tools, and might be reduced through system-level optimizations.
>
> Importantly, although LMLM introduces higher per-token latency, it achieves comparable factuality with *substantially smaller model size* (e.g., LMLM-LLaMA2-382M vs. LLaMA2-7B), which is often a favorable tradeoff in compute or memory constrained settings.
>
> In Section 5 (see Figure 8),  we extend the trade-off analysis between storing knowledge in parameters versus offloading it to the external database. We now include the corresponding compute budgets. As shown below, higher offloading consistently improves language modeling (lower perplexity) and factual precision with a modest increase in compute. We include it in the updated Appendix D and limitation section in Appendix E.
>
> | Method               | Database Construction (FLOPs)        | Training Cost (H100-days) | Training Cost (FLOPs)            | Inference Latency (ms/token) |
> |----------------------|---------------------------------------|----------------------------|----------------------------------|-------------------------------------|
> | Standard Pretraining | N/A                                   | 4.80 (1.0×)                | 3.67 × 10^19 (1.0×)              | 9.53                        |
> | LMLM-382M (Ours)          | 5.51 × 10^19 (one time) | 8.72 (~1.82×)              | 6.32 × 10^19 (~1.72×)            | 21.08 (~2.21×)                      |
>
> > *Note*: The measurements are based on a knowledge-intensive corpus, representing a worst-case estimate of compute budgets.
>
> | Knowledge Offload Ratio | Training GPU Hours (H100-days) ↓ | Negative Log-Likelihood (NLL) ↓ | FactScore ↑ |
> |-------------------------|------------------------------|----------------------------------|-------------|
> | 0% (Standard LLM)†        | ~ 2-3                            | 8.50                             | 10.1        |
> | 25%                     | 2.48                         | 7.71                             | 17.2        |
> | 50%                     | 2.91                         | 7.43                             | 26.1        |
> | 75%                     | 4.10                         | 7.19                             | 27.1        |
> | 90%                     | 4.31                         | 6.98                             | 29.2        |
> | 100% (LMLM)†                   | ~5-6                            | 6.70                             | 30.6        |
>
> > † The 0% and 100% training runs were launched on slightly different GPU configurations, so their training GPU hours are not directly comparable to the others.
>
>
> ---
>
> ### **Q1: Unlearning performance of LMLM vs. RAG-based models**
>
> We now include a detailed comparison between LMLM and RAG-based models, where both methods access a controlled, pre-stored database. To ensure a fair comparison, we limit the model’s knowledge to the external evidence in the context and enforce strict instructions preventing any use of parametric knowledge. We also skip the retrieval step and directly provide the golden supporting document, so the RAG setup reflects its upperbound performance.
>
> ```
> You are an AI assistant that must answer questions using **ONLY** the information explicitly provided in the Knowledge Base.
>
> CRITICAL RULES:
> 1. You may use **only** information stated in the Knowledge Base.
> 2. You must **not** use parametric knowledge, background knowledge, or any information not included in the Knowledge Base.
> 3. If the Knowledge Base contains **no information** or does not include the answer, reply exactly:
> "I do not know based on the provided knowledge."
>
> Knowledge Base:
> {retrieved_docs if retrieved_docs else "The Knowledge Base is empty."}
>
> Question:
> {query}
> ```

---

> ### Author Response · Authors · 2025-11-21
>
> Here are the detailed results for TOFU on llama-3.1-1b-instruct model. Even though RAG preserves model utility, it fails to achieve effective unlearning (p-value < 0.05) by just controlling the external knowledge in the context. This is because RAG cannot remove or hide parametric knowledge stored in the model weights. We include the new result in Appendix C.6.
>
> | Method     | model utility (-) | forget quality (p-value ↑) |
> |------------|--------------------|-----------------------------|
> | Finetuned  | 0.60               | 1.33e-13                    |
> | Retain     | 0.60               | 1.00                        |
> | GradAscent | 0.00               | 1.94e-119                   |
> | GradDiff   | 0.53               | 1.94e-119                   |
> | IdkDPO     | 0.07               | 1.12e-05                    |
> | NPO        | 0.45               | 0.14                        |
> | SimNPO     | 0.58               | 5.01e-100                   |
> | RMU        | 0.58               | 4.87e-10                    |
> | **RAG†**   | 0.50               | 6.54e-13                    |
> | **LMLM**   | 0.55               | 0.31                        |
>
> † *Note*:  RAG can’t really act as an unlearning method. Unlearning aims to have a model that is statistically indistinguishable from one trained only on the Retain Set, which typically requires removing parametric knowledge by finetuning. Therefore, RAG’s “unlearned” behavior simply reflects the model following instructions to ignore or pretend not to know certain facts, rather than the actual knowledge removal.
>
>
>
> ---
>
> ### **Q2: Analysis of lookup failure cases and constrained decoding methods**
>
> > How did you handle the 'failure cases', for example, when the model generated a syntactically infeasible database query or a query that the database cannot answer, and how often did it happen? Or, did you preclude such possibilities by employing a constrained decoding methods?
>
> **Analysis of lookup failure cases.**
>
> Thank you for the question. We first conducted additional analyses to understand how lookup failures occur and how often they happen. From 50 manually inspected T-REx examples, 64% were exact matches. The remaining failures arose from ambiguous entities/relations (6%), missing database entries (10%), multiple matches (2%), or insufficient context in the T-REx completion setting (18%). Overall, the lookup performs reliably, with the remaining errors mostly coming from limited coverage of the database.
>
> Current fallback behavior either returns the top-1 match or outputs “unknown,” and the model is not yet trained to handle these cases robustly. To improve robustness, future work will explore RL training to enable the model to reason, verify, and backtrack when the queried external knowledge is unavailable.
>
> **Constrained decoding.**
>
> As an alternative, we also implemented *prefix-tree constrained decoding* (Appendix A.4, D.3), which restricts generation to syntactically valid database entries and supports beam search over valid prefixes.
>
> | Model | Type | Decoding | FActScore (%) ↑ | FActScore w/o penalty ↑ | Facts / Response ↑ |
> | ----------- | -------- | ----------- | --------------- | ----------------------- | ------------------ |
> | LLaMA2-176M | Standard | - | 10.1 | 11.8 | 23.7 |
> | | | Prefix-tree | 23.0 | 23.9 | 34.6 |
> | | | Fuzzy Match | 30.6 | 32.4 | 31.4 |
> | LLaMA2-382M | LMLM | - | 14.0 | 16.6 | 32.4 |
> | | | Prefix-tree | 23.5 | 31.3 | 28.1 |
> | | | Fuzzy Match | 31.9 | 32.7 | 33.7 |
>
> As shown above, fuzzy matching performs better in open-ended generation on Factscore metric. Prefix-tree decoding helps reduce hallucinations by enforcing validity, but it does so at the cost of flexibility. As both the model and the database grow, we expect structured decoding to become more effective, which also suggests that the main source of lookup failures today is limited database coverage. For future work, we plan to explore RL-based extraction and annotation to improve database coverage. We have added this discussion and limitation to the Appendix D.6.

---

> > ### Comment · Reviewer_qoG2 · 2025-11-26
> >
> > Thank you for the detailed response. The responses and additional experiments have clarified most of my concerns, and I appreciate the authors’ effort to integrate them into the revision. Overall, I believe this is a technically solid work with clear contributions, and I am happy to increase the score by 2. That said, I am still not convinced whether the proposed approach can eventually be scaled and advanced to deal with more complicated syntactic structures in realistic environments and post-training use, which keeps me from further increasing the score.

---

> > > ### Author Response · Authors · 2025-11-26
> > >
> > > Thank you for the thoughtful follow-up and thanks for increasing the score. We appreciate the time you spent reviewing our work.
> > >
> > > Regarding your remaining concern about scalability to more complex syntactic structures and realistic post-training settings, we agree this is an important direction. We have added clearer discussion of these limitations and future extensions in the revised paper. Thanks again for the constructive feedback.

---

### Official Review · Reviewer_s4z7 · 2025-11-01

**Soundness:** 3
**Presentation:** 3
**Contribution:** 3
**Rating:** 6
**Confidence:** 3

**Summary:**

The authors propose Limited/Large Memory Language Models (LMLM), a pre-training framework in which specific factual knowledge is stored in an external database rather than in model parameters. During training, a lightweight annotator marks entity–relation–value triples in the corpus and inserts lookup calls into the text. The retrieved factual values are masked out of the loss, prompting the model to learn when to consult the database instead of memorizing facts. At inference, LMLM generates text and performs lookups as needed, retrieving accurate information from the database. The experiments show that LMLM can match or surpass standard models of similar size in perplexity and factual accuracy, achieve results comparable to much larger models on factual benchmarks, and support efficient unlearning by editing the external knowledge base.

**Strengths:**

* Introducing masking of retrieved factual values during pre-training to encourage reliance on external lookups rather than weight memorization is a compelling idea. It contributes to ongoing discussions around modularizing knowledge in language models.

* The paper presents a complete pipeline, from data annotation to model training and inference, which may facilitate adoption or extension by other practitioners.

**Weaknesses:**

* The method’s success relies heavily on an annotation pipeline using GPT-4o and a trained annotator model to extract factual triples. The manuscript would benefit from deeper analysis of annotation accuracy, coverage, biases introduced by the seed annotations, and scalability to larger or more diverse corpora.

* The focus on (entity, relation → value) triples means the approach externalizes only certain types of knowledge (birth dates, titles, etc.). More complex or contextual knowledge (e.g., procedural knowledge, narratives, or common sense) remains embedded in the model weights. The paper does not explore how well the approach generalizes beyond these atomic facts.

* Fuzzy matching via sentence embeddings and a similarity threshold is pragmatic but may encounter failures (e.g., entity ambiguity, missing or conflicting entries). Have authors done experiments on this?

*  While the authors note that the computational cost of external lookups is non-negligible, a more thorough discussion of latency, particularly in real-time applications, and possible mitigation strategies would be valuable.

* The paper contextualizes its contribution relative to kNN-LM, RAG, and other prior work. However, additional comparisons to recent approaches that incorporate explicit memory or knowledge bases (e.g., Memory, KBLaM, MLP Memory, RET-LLM) could clarify how LMLM advances the field beyond those efforts.


KBLaM: Knowledge Base augmented Language Mode, Wang et. al. 2024

Memory³: Language Modeling with Explicit Memor, Yang et. al., 2024

MLP Memory: A Retriever-Pretrained Memory for Large Language Models, Wei et. al. 2025

RET-LLM: Towards a General Read-Write Memory for Large Language Models, Modarressi et. al. 2024

MemLLM: Finetuning LLMs to Use An Explicit Read-Write Memory, Modarressi et. al. 2024

**Questions:**

* Have the authors done any comparison with the same model trained on the same corpus but without the knowledge removal?

---

> ### Author Response · Authors · 2025-11-21
>
> Thanks for your thoughtful review. Your feedback is very helpful in strengthening the clarity and completeness of the paper. We address each concern below:
>
> ---
> ### **W1: Further analysis of annotation accuracy, coverage, biases introduced by the seed annotations, and scalability to larger or more diverse corpus.**
>
> Thank you for raising this point. LMLM requires a one-time annotation to build the database and insert the lookup calls. This corresponds to a single inference pass of the annotator model (LLaMA-3.1-8B-Instruct) over the Wikipedia corpus, costing roughly 5 × 10¹⁹ FLOPs (within two days on 64×A6000 GPUs). This one-time preprocessing cost is comparable to pretraining a ~300M model on ~30B tokens, and can be further reduced by distilling a smaller annotator or distributed across multiple clusters.
>
> **Annotation accuracy and coverage:**
> To understand annotation quality, we manually inspect 50 T-REx examples: 64% were exact matches. The remaining failures arose from ambiguous entities/relations (6%), missing database entries (10%), multiple matches (2%), or insufficient context in the T-REx completion setting (18%). Overall, the annotator performs reliably, with the remaining errors mostly coming from limited coverage and the representational limitation of triplets, rather than the annotator generating incorrect or hallucinated lookup. For future work, we plan to extend the annotation framework to cover more complex or contextual knowledge and improve robustness through RL-based extraction and annotation to improve database coverage.
>
> **Bias and potential risks:**
> We are also aware of the potential risks and bias of introducing annotations directly into pretraining data. Additional filtering and safeguards will be important for mitigating safety-related or malicious-content risks.
>
> **Scalability:**
> The current implementation focuses on entity-level factual knowledge, which captures only a subset of the broader knowledge spectrum. Scaling to larger or more diverse corpora will require methods capable of extracting and offloading more complex forms of knowledge, as discussed in the limitation in sec 6. We view this as an important direction for future work.
>
> We have added this discussion and limitation to the Appendix E.
>
> ---
> ### **W2: More complex or contextual knowledge (e.g., procedural knowledge, narratives, or common sense) remains embedded in the model weights.**
>
> Our work explores whether LLMs can be developed to explicitly disentangle certain types of knowledge to non-parametric sources. Since our focus was on demonstrating the effectiveness of this novel approach, we deliberately chose to work within a well-defined category, entity-level factual knowledge, to ensure clarity and rigor in evaluation. This choice was further motivated by prior work showing LLMs struggle with long-tail entities[1], making it a tractable and meaningful starting point. Our findings show that this form of knowledge can be externalized without degrading performance in general language understanding (sec 4.2), while enabling benefits like instant unlearning (sec 4.3), which are difficult to support with fully parametric models.  As discussed in W1, our qualitative analysis confirms that knowledge annotation and targeted externalization on entity-level factual knowledge work reliably within this scope, strengthening the validity of our results.
> Therefore, we view this work as the initial demonstration that this kind of disentanglement is even possible. We are excited to extend LMLM paradigm to more complex or contextual knowledge.
>
> [1] Kandpal, Nikhil, et al. "Large language models struggle to learn long-tail knowledge." ICML 2023.

---

> ### Author Response · Authors · 2025-11-21
>
> ### **W3: More experiments addressing failure cases of fuzzy matching**
>
> Thank you for the question. In our setup, the model generates lookup queries that are matched to the database using cosine similarity over sentence embeddings. While this *fuzzy matching* offers flexibility, it can occasionally yield ambiguous or syntactically invalid queries, leading to incorrect retrieval.
>
> As an alternative, we also implemented *prefix-tree constrained decoding* (Appendix A.4, D.3), which restricts generation to syntactically valid database entries and supports beam search over valid prefixes.
>
> | Model | Type | Decoding | FActScore (%) ↑ | FActScore w/o penalty ↑ | Facts / Response ↑ |
> | ----------- | -------- | ----------- | --------------- | ----------------------- | ------------------ |
> | LLaMA2-176M | Standard | - | 10.1 | 11.8 | 23.7 |
> | | | Prefix-tree | 23.0 | 23.9 | 34.6 |
> | | | Fuzzy Match | 30.6 | 32.4 | 31.4 |
> | LLaMA2-382M | LMLM | - | 14.0 | 16.6 | 32.4 |
> | | | Prefix-tree | 23.5 | 31.3 | 28.1 |
> | | | Fuzzy Match | 31.9 | 32.7 | 33.7 |
>
> As shown above, fuzzy matching performs better in open-ended generation on Factscore metric. Prefix-tree decoding reduces hallucinations by enforcing validity, but it limits flexibility. As both the model and the database grow, we expect structured decoding to become more effective. For future work, we plan to explore RL-based extraction and annotation to improve database coverage. We have added this discussion and limitation to the Appendix E.
>
>
>
> ---
> ### **W4: Details on inference latency and possible mitigation strategies**
>
> We report inference costs under the same decoding setup.  At inference, generating and executing lookup calls introduces a latency increase of 2.21×, which we have not yet optimized. This overhead is similar to that observed in current agentic LLMs that call external tools, and might be reduced through system-level optimizations.
>
> Importantly, although LMLM introduces higher per-token latency, it achieves comparable factuality with *substantially smaller model size* (e.g., LMLM-LLaMA2-382M vs. LLaMA2-7B), which is often a favorable tradeoff in compute or memory constrained settings.
>
> In latency-sensitive or real-time settings, several mitigation strategies are possible:
>
> (1) Instead of generating natural-language entity and relation phrases, the model can emit a single compact knowledge token that maps directly to an entry in the database, reducing both generation length and lookup time.
>
> (2) Lookup results can be cached across sentences, or user turns. We include these strategies as future directions.
>
> | Method         | Avg Total Tokens  / Answer      | Avg Valid Tokens / Answer           | Latency (ms/token)            |
> |----------------|---------------------------|------------------------------|-------------------------------|
> | Standard       | 214.49                    | 214.49                       | 9.53                          |
> | LMLM (Ours)    | 399.95 (1.86×)        | 206.58 (0.96×)          | 21.08 (2.21×)             |
>
>
> > *Note*: The measurements are based on a knowledge-intensive corpus, representing a worst-case estimate of compute budgets.

---

> > ### Author Response · Authors · 2025-11-21
> >
> > ### **W5: Related works (e.g., Memory, KBLaM, MLP-Memory, RET-LLM)**
> >
> > Thanks for your suggestion. We have expanded the related work section and added a detailed comparison in the table below (full version in Table 13 in the appendix).
> >
> > Specifically, KBLaM augments knowledge in the form of continuous key-value vectors as knowledge tokens and integrates them into pre-trained LLMs with a modified rectangular attention structure.
> > MemLLM (formerly RET-LLM) proposes an explicit read-write memory, enabling LLM to extract knowledge into triplets, stores and later recalls knowledge for specific tasks.
> > Concurrently, MLP-Memory distills a kNN retriever into a lightweight MLP without accessing documents, which improves factuality and inference throughput.
> > Memory^3 pretrains models to write and read sparse attention key-values as explicit memories, reducing the burden of model parameters to memorize specific knowledge.
> >
> > KBLaM, MemLLM, MLP-Memory focus on lightweight explicit memory and integrate it via architectural modification or finetuning. Memory^3 is close to our LMLM in that both pretrain models from scratch to access external memory to reduce parametric memorization, however, LMLM enjoys the benefits of an external natural-language knowledge database, which is easy to update, verify and unlearn. In addition, LMLM offers a conceptually simple training recipe that can be integrated by finetuning existing LMs on knowledge-intensive domains.
> >
> > | Model | Architecture | Internal Storage | External Storage | Integration | PPL ↓ | Factuality ↑ | Unlearning | Goal |
> > |-------|--------------|------------------|------------------|-------------|-------|--------------|------------|------|
> > | KBLaM      | Dec-only | ✓         | Continuous KV     | Modified rectangular attention | ✗     | ✓         | ✗          | Integrate KB tokens                    |
> > | MemLLM     | Dec-only | ✓         | Extracted triplets | Read-write calls             | ✓     | ✓         | ~          | FT to use read-and-write memory        |
> > | MLP-Memory | Dec-only | ✓         | MLP-based         | MLP imitates kNN retrieval    | ✓     | ✓         | ✗          | High-throughput memory                 |
> > | Memory³    | Dec-only | ✓         | Sparse KV         | Self-attention               | ✓     | ✓         | ✗          | Reduce parametric memorization         |
> > | LMLM (Ours) | Dec-only | ✓ (partial) | KB | Explicit lookup | ✓ | ✓ | ✓ | Decouple facts from weights |
> > > Note: ✓ = supported, ✗ = not supported, ~ = partial support (e.g., edit without full unlearning or not explicitly evaluated).
> >
> >
> >
> >
> > ---
> >
> > ### **Q1: Comparison with the same models trained on the same corpus but without the knowledge removal**
> >
> > We performed ablation study using the same model architecture trained on the same corpus under the same training setup, but without (a) lookup calls and/or (b) the masked loss. We report the eval perplexity below.
> >
> > | Model Type | Model | Dynamic PPL ↓ | Normalized PPL ↓ | Static PPL ↓ |
> > |----------------|--------|-----------|--------------|-----------|
> > | LMLM | LLaMA2-176M | **6.69** | **6.69** | **5.84** |
> > | - w/o masked loss | LLaMA2-176M | 6.81 | 6.82 | 5.85 |
> > | - w/o lookup and masked loss | LLaMA2-176M | – | – | 8.52 |
> > > *Note*: - w/o lookup and masked loss corresponds to the Standard LLM baseline in the paper.
> >
> > These results show that both (1) learning to perform database lookups and (2) applying the masked loss on return values improve pretraining efficiency. Removing the masked loss causes the model to revert toward parametric memorization, leading to **slightly worse perplexity**.
> >
> > More importantly, the masked-loss component is responsible for decoupling factual memorization from learning linguistic competency. This effect is shown more predominantly in the unlearning setting. Using the same setting, we train an ablation version w/o the masked loss and eval on the TOFU benchmark.
> >
> >  | Method     | model utility (-) | forget quality (p-value ↑) |
> > |------------|--------------------|-----------------------------|
> > | Finetuned  | 0.60               | 1.33e-13                    |
> > | Retain (Ideal)     | 0.60               | 1.00                        |
> > | **LMLM (Ours)**   | 0.55               | 0.31                        |
> > | **LMLM w/o masked loss**   | 0.56               | 0.01                        |
> >
> > The ablation without the masked loss **fails to unlearn** targeted facts (p-value ≈ 0.01), confirming that the masked loss effectively discourages LMLM from memorizing facts. This decoupling forms the core advantage of LMLM over the knowledge-dense counterparts. It enables *instant unlearning* simply by editing the external database, which is difficult to achieve with post-hoc methods.
> > We highlight these ablations in the updated Appendix C.7.

---

### Official Review · Reviewer_7MFj · 2025-11-01

**Soundness:** 2
**Presentation:** 3
**Contribution:** 4
**Rating:** 8
**Confidence:** 4

**Summary:**

The paper proposes a novel pre-training paradigm, which aims to remove knowledge from the pre-training process and requires most parameters to only learn capabilities.

The paper’s specific approach is to label the knowledge pairs in all pre-training corpora and mask the knowledge, thereby suppressing knowledge memorization. Among this process, the labeling for pre-training is completed by a small model obtained through distillation.

**Strengths:**

- The 355M small model trained by the authors has achieved better results on metrics such as FactScore, even when compared to similar models augmented with RAG.
- The training paradigm proposed by the authors is highly innovative, radical yet cost-effective. This is because the cost of searching is far lower than that of memorizing knowledge using large amounts of pre-training data.

**Weaknesses:**

- For a pre-training paradigm, it is clearly unreasonable for the authors to evaluate and compare models of the same scale solely using factuality-related evaluation methods. I believe the authors should at least add evaluations on aspects representing instruction following and reasoning capabilities, though I understand this is difficult for an ultra-small-scale model.
- The authors’ training scale is too small to allow us to determine whether the advantages currently achieved can be overwritten in subsequent training phases.
- The annotation cost for each piece of data is relatively high, and there is a certain error rate. While I am aware that many datasets also use processing methods with similar costs, those methods are only used for tasks like quality scoring and adjusting training volume. The risks associated with the act of directly adjusting masks do not seem to have been fully discussed.

**Questions:**

Can models trained through this method tackle tasks that require complex background knowledge and theories, such as math problems?

If I train a capability-only model, does this mean I need to incorporate an entire math textbook into the prompt to provide the corresponding knowledge when facing such tasks?

Beyond that, a more thought-provoking point is: Have the authors proven that the model’s memory occupies a large number of parameters, leading to an insufficient proportion of capability-related parameters and thus deficiencies in capabilities? I believe this may not have been proven.

In fact, as open-source models continue to advance, even small models like Qwen3-4B can hold their own in certain capabilities such as math reasoning, while performing much worse in terms of knowledge. Does this imply that as the quality of training data improves, models store knowledge and capabilities with higher compression efficiency? If so, the authors’ work may yield limited results.

---

> ### Author Response · Authors · 2025-11-21
>
> Thanks for your thoughtful review and for recognizing the novelty and effectiveness of our pretraining paradigm. We appreciate your clear understanding of the scope of this work. Below, we respond to your concerns.
>
> ---
>
> ### **W1: Evaluation beyond factuality (instruction-following and reasoning capabilities) for pretraining paradigm**
>
> We agree that evaluating instruction-following or reasoning ability is important for understanding a pretraining paradigm. At the small scale, however, it is challenging to meaningfully evaluate later-stage capabilities (e.g. instruction following or RL-based reasoning), which is a limitation we acknowledge.
>
> Among current open-source small models (e.g., Pythia, Olmo, TinyLlama, SmolLM3) [1-4], only SmolLM3-3B performs mid-training and post-training evaluation at scale. It remains standard practice for base models to be evaluated through pretraining NLU tasks and perplexity [5, 6], which serve as reliable indicators for data quality and pretraining effectiveness. Within NLU, HellaSwag and CommonsenseQA emphasize commonsense reasoning, while PIQA includes knowledge and linguistic understanding. We include these in Table 12 and report perplexity in Figure 5.
>
> We agree that it remains an exciting and open question how to design base models that are inherently RL-able [7] or naturally capable of tool use. Within the scope of this paper, we focus on knowledge memorization during pretraining. We leave it for future work and have included this discussion to the updated appendix E.
>
> [1] Biderman, Stella, et al. "Pythia: A suite for analyzing large language models across training and scaling." International Conference on Machine Learning. PMLR, 2023.
>
> [2] Groeneveld, Dirk, et al. "Olmo: Accelerating the science of language models." Proceedings of the 62nd annual meeting of the association for computational linguistics (volume 1: Long papers). 2024.
>
> [3] Zhang, Peiyuan, et al. "Tinyllama: An open-source small language model." arXiv preprint arXiv:2401.02385 (2024).
>
> [4] Bakouch, Elie, et al. "SmolLM3: smol, multilingual, long-context reasoner." Hugging Face Blog (2025).
>
> [5] Magnusson, Ian, et al. "Datadecide: How to predict best pretraining data with small experiments." ICML, 2025.
>
> [6] Soldaini, Luca, et al. "Dolma: An open corpus of three trillion tokens for language model pretraining research." Proceedings of the 62nd annual meeting of the association for computational linguistics (volume 1: long papers). 2024.
>
> [7] Yang, An, et al. "Qwen3 technical report." arXiv preprint arXiv:2505.09388 (2025).
>
> ---
> ### **W2: Whether the pretraining advantages can persist at larger scales or under further training**
>
> We agree that evaluating whether the observed advantages persist at larger scales and under continued training is important for understanding the generality of this paradigm. First, prior work showing LLMs, from small (560M) to moderately large scale (176B), all struggle with learning long-tail entities during pretraining [1]. This difficulty appears to be inherent to standard language modeling, which motivates LMLM to explicitly offload entity-level factual knowledge to external database rather than memorization.
>
> Our mechanism operates purely through data-level supervision. In principle, it can be made compatible with instruction tuning by consistently applying the same LMLM-style entity-level annotation and masked next token prediction loss to knowledge-intensive instruction data, while leaving other instruction data unchanged.
>
> More importantly, instruction tuning mainly teaches models to respond to diverse end tasks and user interactions, exposing the knowledge and capabilities learned during pretraining [2-3]. It aligns response behavior rather than acquiring new knowledge, while LMLM controls whether factual knowledge is offloaded and stored externally. These objectives are complementary.
>
> Other aspects (e.g. reasoning or tool-use ability) remain open questions as discussed in W1. We have added this discussion to the updated Limitations section.
>
>
> [1] Kandpal, Nikhil, et al. "Large language models struggle to learn long-tail knowledge." ICML 2023.
>
> [2] Zhou, Chunting, et al. "Lima: Less is more for alignment." Advances in Neural Information Processing Systems 36 (2023): 55006-55021.
>
> [3] Sanh, Victor, et al. "Multitask prompted training enables zero-shot task generalization." arXiv preprint arXiv:2110.08207 (2021).

---

> > ### Author Response · Authors · 2025-11-21
> >
> > ### **W3: Cost, error rate and potential risk of the annotation process**
> >
> > Thank you for raising this point. LMLM requires a one-time annotation to build the database and insert the lookup calls. This corresponds to a single inference pass of the annotator model (LLaMA-3.1-8B-Instruct) over the Wikipedia corpus, costing roughly 5 × 10¹⁹ FLOPs (within two days on 64×A6000 GPUs). This one-time preprocessing cost is comparable to pretraining a ~300M model on ~30B tokens, but it can be further reduced by distilling a smaller annotator or distributed across multiple clusters.
> >
> > To understand annotation quality, we manually inspect 50 T-REx examples: 64% were exact matches. The remaining failures arose from ambiguous entities/relations (6%), missing database entries (10%), multiple matches (2%), or insufficient context in the T-REx completion setting (18%). Overall, the annotator performs reliably, with the remaining errors mostly coming from limited coverage and the representational limitation of triplets, rather than the annotator generating incorrect or hallucinated lookup. For future work, we plan to extend the annotation framework to cover more complex or contextual knowledge and improve robustness through RL-based extraction and annotation to improve database coverage.
> >
> > We are also aware of the potential risks of introducing annotations directly into pretraining data. Additional filtering and safeguards will be important for mitigating safety-related or malicious-content risks. We have added this discussion and limitation to the Appendix E.
> >
> >
> >
> >
> >
> > ---
> > ### **Q1: How capability-only models handle math problems**
> > > If I train a capability-only model, does this mean I need to incorporate an entire math textbook into the prompt?
> >
> > Yes and no. This depends on how internal knowledge and external knowledge are defined when building capability-only models.
> >
> > Knowledge spans a broad spectrum from specific entity-level facts to more general knowledge (e.g. math theorem or reasoning patterns). Our work focuses on offloading entity-level facts, which is often long-tailed and difficult for LLMs to memorize [1]. Such knowledge is naturally externalized to a database. In contrast, mathematical reasoning is structurally different in that models benefit from learning diverse reasoning patterns rather than memorizing isolated facts. How to construct a datastore that enables retrieval of such patterns remains an open challenge [2]. For this reason, in capability-only models we do not attempt to externalize such reasoning knowledge. We believe that general reasoning knowledge should remain part of the model’s internal capability, which the model can recall and apply directly when solving math problems.
> >
> >
> >
> >
> > ---
> > ### **Q2: Does knowledge memorization occupy too many parameters?**
> >
> > > “Have the authors proven that the model’s memory occupies a large number of parameters, leading to an insufficient proportion of capability-related parameters and thus deficiencies in capabilities?”
> >
> > We agree that the field does not have a precise decomposition of “knowledge parameters” vs. “capability parameters.” Our intuition is based on several converging empirical findings from prior work:
> >
> > (1) FFN layers act as key–value memories for factual associations [4], and causal-editing methods (e.g., ROME [5]) show that modifying a small subset of FFN parameters reliably changes specific facts. Since FFNs constitute roughly two thirds of model parameters, this suggests that factual knowledge occupies a substantial portion of model capacity.
> >
> > (2) Scaling-law studies show that knowledge tasks are more **capacity-hungry** than coding tasks at fixed compute [6], implying that factual memorization competes more directly for parameters.
> >
> > (3) Prior work shows strong correlation between model size and factual memorization [7, 1]. However, factual knowledge is long-tailed and rarely repeated [3], making parametric storage inefficient compared to an external store.
> >
> > These findings provide consistent evidence that LLMs consume substantial capacity to encoding factual knowledge, especially long-tail knowledge. This motivates us to externalize such factual knowledge into an external database rather than storing it in parameters. LMLMs offer a pathway toward language models with substantially reduced dependency on parameter count for factual accuracy, potentially shifting how future language
> > models store, access, and maintain knowledge

---

> > > ### Author Response · Authors · 2025-11-21
> > >
> > > ### **Q3: Do improved data quality limit the benefits of the proposed method?**
> > > > As small open-source models (e.g., Qwen3-4B) show good reasoning but weak factuality, does improved data quality make capability more compressible?
> > >
> > > This is an interesting question. While some formulations of factual sentences may be more compressible than others, improving data quality provides limited benefit for factual knowledge memorization. The key bottleneck is not sentence quality but **frequency**: prior work shows that models typically need to observe a fact hundreds to thousands of times to memorize it reliably [3]. Entity-level facts, however, are highly long-tailed and thus rarely repeated in pretraining corpora [2]. This limitation is inherent to standard language modeling and cannot be fully addressed by higher-quality data alone. Therefore, it is natural to externalize such long-tail factual knowledge into a database, as proposed in LMLM.
> > >
> > > Regarding the Qwen example, it is impressive that such a small model can already exhibit strong reasoning capabilities. Better-curated and more diverse data can meaningfully improve reasoning capabilities, but this does not solve the challenges of factuality or knowledge controllability.
> > >
> > > Importantly, our approach is complementary to efforts such as improved data quality or enhanced reasoning-oriented pretraining. LMLM specifically targets knowledge controllability and verification, which high-quality data alone cannot resolve easily.
> > >
> > >
> > > ---
> > > #### Reference
> > >
> > > [1] Kandpal, Nikhil, et al. "Large language models struggle to learn long-tail knowledge." ICML 2023.
> > >
> > > [2] Lyu, Xinxi, et al. "Frustratingly Simple Retrieval Improves Challenging, Reasoning-Intensive Benchmarks." arXiv preprint arXiv:2507.01297 (2025).
> > >
> > > [3] Allen-Zhu, Zeyuan, and Yuanzhi Li. "Physics of language models: Part 3.3, knowledge capacity scaling laws." ICLR 2025.
> > >
> > > [4] Geva, Mor, et al. "Transformer feed-forward layers are key-value memories." Proceedings of the 2021 Conference on Empirical Methods in Natural Language Processing. 2021.
> > >
> > > [5] Meng, Kevin, et al. "Locating and editing factual associations in gpt." Advances in neural information processing systems 35 (2022): 17359-17372.
> > >
> > > [6] Roberts, Nicholas, et al. "Compute optimal scaling of skills: Knowledge vs reasoning." Findings of the Association for Computational Linguistics: ACL 2025. 2025.
> > >
> > > [7] Morris, John X., et al. "How much do language models memorize?." arXiv preprint arXiv:2505.24832 (2025).

---

> > > > ### Comment · Reviewer_7MFj · 2025-11-26
> > > >
> > > > I appreciate the authors' detailed response.
> > > >
> > > > I stand by my original assessment—the novelty and inspiration of this work are sufficiently remarkable that certain limitations can be overlooked.
> > > >
> > > > I will maintain my high rating.

---

> > > > > ### Author Response · Authors · 2025-11-26
> > > > >
> > > > > Thank you for the thoughtful review and for maintaining the high rating. We truly appreciate your time and your encouraging comments on the novelty and impact of this work.

---

### Author Response · Authors · 2025-12-03

**Dear AC and Reviewers**,

We sincerely thank the reviewers for their thoughtful feedback and the ACs for handling our submission.

Our paper proposed Limited Memory Language Models (LMLM), a new class of language models that externalizes factual knowledge to external database during pre-training rather than memorizing them. Our experiments show that LMLMs achieve competitive performance compared to significantly larger LLMs on standard benchmarks, while offering the advantages of explicit, editable, and verifiable knowledge bases.

> Note: We use R1–R4 following the reviewer order shown on the ICLR submission page (R1=Reviewer 7MFj, R2=Reviewer s4z7, R3=Reviewer qoG2, R4=Reviewer pSTT).

We appreciate the reviewers’ recognition of the **novelty** of our approach (R1, R4), the substantial implementation effort (R2, R4), the **comprehensive evaluation and strong results** (R1, R3, R4), and the **clear limitations discussion** (R3). We are also encouraged by reviewers’ comments that this work **contributes meaningfully to the broader direction of modularizing knowledge in language models** (R2).

During rebuttal and discussion, we added new experiments and clarifications addressing all major concerns, including:
- New results and further analyse
    - Knowledge-editing benchmarks (e.g., MQuake) (R4)
    - Annotation cost, training/inference cost, and safety considerations (R1, R2, R3)
    - Lookup failure analysis and potential constrained decoding strategies (R2, R3)
- Additional ablations and baselines
    - Annotation pipeline and pretraining ablation (R2, R4), unlearning performance compared to RAG (R3)
- Expanded Related work (R2)
- More detailed limitations and future work section in the updated Appendix E

During the discussion, we appreciated the early responses from reviewers R1, R3, and R4.
- R1 and R4 **maintain scores of 8**, noting that this is a strong and inspiring paper (R4), and that the novelty justifies overlooking certain limitations at this stage (R1)
- R3 states that this is a technically solid work with clear contributions and **increased the score to 6**
- R2 has not posted a response yet with original score of 6

Overall, we are encouraged by the positive consensus. Our revision incorporates all requested new experiments, clarifications, and expanded discussions of limitations and future work, and we believe the paper is now significantly strengthened.

---

### Meta-Review · Area_Chair_VjiL · 2026-01-13

**Summary:**

This paper proposes to use a pre-trained LLM (4o) to annotate and extract factual knowledge during the pre-training phase of another LLM. This knowledge is transferred to an external database that can be looked up as a tool during inference. The basic idea is somewhat obvious and not new. Reviewers generally appreciated the detailed experiments and results.

**Reviewer Concerns:**

- 7MFj Evaluation beyond factuality: **No**
- 7MFj Do advantages persist: **No**
- 7MFj Cost, error rate and potential risk: Yes
- 7MFj Math problems: **No**
- 7MFj Too many parameters: **No**
- 7MFj Improved data quality: Yes
- s4z7 Further analysis: Yes
- s4z7 More complex or contextual knowledge: **No**
- s4z7 More experiments: Yes
- s4z7 Details on inference latency: **No**
- s4z7 Related works: Yes
- s4z7 Comparison with the same models: **No**
- qoG2 Deployment: Yes
- qoG2 Brittleness: Yes
- qoG2 Computational cost: Yes
- qoG2 Unlearning performance: Yes
- qoG2 Lookup failure cases: Yes
- pSTT Additional results: Yes
- pSTT Annotation pipeline ablation: Yes
- pSTT Limitations: Yes
- pSTT Database setup: Yes
- pSTT TOFU setup: Yes

**Reviewer Scores:**

7MFj 8->8
s4z7 6->6
qoG2 4->6
pSTT 8->8

---

### Decision · Program_Chairs · 2026-01-26

Accept (Poster)